# Thermal sensitivity of soil microbial carbon use efficiency across forest biomes

Chengjie Ren [1,2], Zhenghu Zhou [3] ✉, Manuel Delgado-Baquerizo [4], Felipe Bastida [5], Fazhu Zhao [6], Yuanhe Yang [7], Shuohong Zhang[1,2], Jieying Wang [6], Chao Zhang[8], Xinhui Han [1,2], Jun Wang [6], Gaihe Yang [1,2] ✉ & Gehong Wei [9] ✉

Understanding the large-scale pattern of soil microbial carbon use efficiency (CUE) and its temperature sensitivity ($CUE_T$) is critical for understanding soil carbon–climate feedback. We used the $^{18}O$-$H_2O$ tracer method to quantify CUE and $CUE_T$ along a north-south forest transect. Climate was the primary factor that affected CUE and $CUE_T$, predominantly through direct pathways, then by altering soil properties, carbon fractions, microbial structure and functions. Negative $CUE_T$ (CUE decreases with measuring temperature) in cold forests (mean annual temperature lower than 10 °C) and positive $CUE_T$ (CUE increases with measuring temperature) in warm forests (mean annual temperature greater than 10 °C) suggest that microbial CUE optimally operates at their adapted temperature. Overall, the plasticity of microbial CUE and its temperature sensitivity alter the feedback of soil carbon to climate warming; that is, a climate-adaptive microbial community has the capacity to reduce carbon loss from soil matrices under corresponding favorable climate conditions.

Soils are the largest repository of carbon in the terrestrial biosphere[1], which represents 25% of the potential of natural climate solutions[2]. Microbial CUE is the result of carbon taken up by microbes allocating to respiration and to growth, potentially forming biomass and subsequently necromass, that could contribute to soil organic matter accumulation[3,4]. On the one hand, a higher microbial CUE suggests a stronger ability to store soil organic carbon (SOC) due to increased biomass synthesis and the availability of microbial residues for organic matter stability. This is in accordance with the frameworks of microbial efficiency-matrix stabilization[5] and the microbial carbon pump[6]. A recent study demonstrated that microbial CUE is the determining factor for SOC storage and its geographical variation across the world[7]. On the other hand, a high microbial CUE can promote SOC losses via increased microbial biomass and subsequent activities of extracellular enzymes, thus enhancing SOC decomposition[8]. Additionally, from a methodology standpoint, microbial CUE is often estimated by a substrate-dependent approach, i.e., the incorporation and utilization of specific $^{13}C$-labeled substrate (e.g., glucose, carbohydrate, leucine additions, and carbon compound mixtures). However, this approach defines microbial CUE within the constraint of a selected carbon substrate, which cannot reflect the microbial CUE derived from organic compounds widely present in the soil[4]. Alternatively, a substrate-

[1]State key Laboratory for Crop Stress Resistance and High-Efficiency Production, College of Agronomy, Northwest A&F University, Yangling, Shaanxi, China. [2]The Research Center of Recycle Agricultural Engineering and Technology of Shaanxi Province, Yangling, Shaanxi, China. [3]School of ecology, Northeast Asia Biodiversity Research Center, Northeast Forestry University, Harbin, Heilongjiang, China. [4]Laboratorio de Biodiversidad y Funcionamiento Ecosistémico. Instituto de Recursos Naturales y Agrobiología de Sevilla (IRNAS), CSIC, Av. Reina Mercedes 10, Sevilla, Spain. [5]CEBAS-CSIC, Department of Soil and Water Conservation, Campus Universitario de Espinardo, Murcia, Spain. [6]Shaanxi Key Laboratory of Earth Surface System and Environmental Carrying Capacity, Northwest University, Xi'an, Shaanxi, China. [7]State Key Laboratory of Vegetation and Environmental Change, Institute of Botany, Chinese Academy of Sciences, Beijing, China. [8]State Key Laboratory of Soil Erosion and Dryland Farming on the Loess Plateau, Northwest A&F University, Yangling, Shaanxi, China. [9]State key Laboratory for Crop Stress Resistance and High-Efficiency Production, College of Life Sciences, Northwest A&F University, Yangling, Shaanxi, China. ✉e-mail: zhouzhenghuzzh@163.com; ygh@nwsuaf.edu.cn; weigehong@nwsuaf.edu.cn

independent approach incorporating $^{18}$O-$H_2O$ into DNA are employed to sensitively estimate microbial CUE[9,10]. However, there are limited observations available to explore the geographic variance of substrate-independent microbial CUE and their impacts on SOC storage, which subsequently limited our prediction of how changes in microbial processes lead to a net positive or negative feedback for carbon emission.

It has been demonstrated that incorporating temperature-sensitive microbial CUE into models can significantly improve the accuracy of predicting global SOC distribution[11]. Soil ecological theory posits that microbial respiration exhibits a more positive response to temperature than biomass production, thereby presenting an inverse temperature-efficiency relationship[4,8]. This linear decrease in microbial CUE with increasing temperature, defined as the temperature sensitivity of microbial CUE (CUE$_T$), has been used to parameterize SOC dynamics in Earth System Models[8]. However, existing experimental studies have not reported robust patterns; they have shown varied responses such as positive, negative, or no change with temperature based on the $^{13}$C-labeled substrate approach[4,8,12–14]. Microbial CUE is responsible for allocating the assimilated carbon into growth and respiration. Several studies have quantified the temperature sensitivity of respiration at local[15,16], regional[12,17], and global scales[18,19], a greater temperature sensitivity of respiration is suggested in soils with cold climates, low carbon-quality, coarse texture, weak mineral protection, and high microbial activity[20]. Despite all the research conducted thus far, our understanding of the thermal sensitivities of microbial carbon uptake, growth, and CUE substantially lags that of respiration. Large-scale studies that concurrently consider a wide range of multiple environmental factors (climate, soil properties, microbial attributes, carbon biochemical recalcitrance) could not only enable us to quantify the biogeography of microbial CUE$_T$, but also provide the robust microbial physiological parameters necessary for Earth System Models.

Here, we used the $^{18}$O-$H_2O$ tracer method at six measuring temperatures (5, 10, 15, 20, 25, and 30°C) to assess soil microbial CUE and CUE$_T$ across a 3425-km north-south forest transect in China (spanning approximately 27 latitudinal degrees; see Supplementary Table 1). As potential drivers of microbial CUE and CUE$_T$, the direct and indirect effects of climate, soil properties, carbon quality, microbial community structure, and the functional genes involved in carbon decomposition were evaluated using structural equation modeling. Climate factors included mean annual temperature (MAT) and mean annual precipitation, with ranges from 3.1 to 23.2°C and 486 to 2266 mm, respectively (Supplementary Table 1). Soil properties examined included soil pH, bulk density, and texture (Supplementary Table 2). Solid-state $^{13}$C cross polarization-magic angle spinning nuclear magnetic resonance spectroscopy and acid hydrolysis methods were utilized to quantify soil carbon quality. High carbon quality (desirable for microbes) is indicative of low molecular weight and structural complexity, but high solubility and lability. The microbial community structure was represented by microbial diversity (Shannon index), fungal abundance, bacterial abundance, the ratio of fungi to bacteria, and the relative abundance of microbial phyla. Metagenomic sequencing was employed to explore the abundance of functional genes associated with the decomposition of various forms of SOC. The objectives of our study were to identify how microbial CUE and its thermal sensitivity change across forest biomes from tropical to temperate regions, ascertain the potential drivers of these changes and evaluate the implications for the future development of soil carbon models.

## Results and discussion
### Latitudinal gradient of microbial physiology
Mass-specific microbial growth (microbial growth per unit microbial biomass carbon) decreased with increasing MAT, despite variations in the measuring temperature (Fig. 1a; Supplementary Table 3). This

pattern aligns with the latitudinal compensation hypothesis in macro-ecology, namely, organisms in cold environments increase their basal metabolic rates and have high potential growth rates to compensate for brief growing seasons[21–23]. In contrast to microbial growth, trends of mass-specific respiration (microbial respiration per unit microbial biomass carbon) along MAT varied with measuring temperatures (Fig. 1b; Supplementary Table 3). Microbial CUE ranged from 0.28 to 0.77 across forests and incubation temperatures (Fig. 1c), concurrent with prior estimates (ranging from near 0 to over 0.8)[10,24]. These broad variations underscore the need for further research into the mechanisms influencing soil microbial CUE.

The influence of environmental variables on soil microbial CUE was examined using Pearson's correlation analysis (Fig. 2) and structural equation modeling (Fig. 3; and Supplementary Fig. 1). The combined effect of climate, soil properties, carbon quality, microbial community structure, and functional genes involved in carbon decomposition accounted for 87% of the variance in microbial CUE (Fig. 3a; and Supplementary Fig. 1). Climate was the primary factor and adversely affected microbial CUE primarily through a direct pathway (Fig. 3a; and Supplementary Fig. 1). Applying the same substrate-independent method but with varying measuring temperatures (mean growing season temperature), Wang et al. documented the negative correlation between microbial CUE and MAT[24], attributing this occurrence to climate-induced changes in plant-soil-microbial properties. Previous studies have shown that climate can alter the chemistry of plant litter and root exudates, which subsequently influence carbon availability for microbial growths, thereby affecting microbial CUE[5,25]. Here, we discovered that both microbial CUE and microbial growth consistently fell as MAT increased across different measuring temperatures (Fig. 1c; Supplementary Table 3). In addition, microbial CUE was more related to growth rather than to respiration (Supplementary Figs. 2 and 3). These results together suggest that the reduction in microbial CUE with MAT was largely dependent on microbial growth rather than respiration.

Climate also regulates microbial CUE by altering the microbial community structure (Fig. 3a). We found a consistently positive association between the fungi to bacteria ratio and microbial CUE among different measuring temperatures (Fig. 2). It is not surprising that fungal-dominated communities are better adapted to low-temperature and low carbon quality conditions (Supplementary Table 2, Supplementary Figs. 4 and 5) and can retain more carbon in biomass per unit of substrate consumed, releasing less as $CO_2$ than bacteria-dominated communities[26,27]. However, we found that microbes fed with low-quality carbon (i.e., greater alkyl, carboxy, and recalcitrant carbon) that encoded recalcitrant carbon-degrading genes (lipids and lignin) had a greater microbial CUE (Fig. 2), which does not support the theoretical assumption that complex substrates require more energy (supported by respiration) to invest in enzyme production and excretion before they can be utilized by the microbial community[4,5]. In accordance with this pattern, a global synthesis found that microbial communities using complex substrates (including cellulose, plants, and microbial cell walls) have comparable or even higher CUE than those using labile glucose[28]. It is worth noting that functional genes from metagenomic sequencing may not equal the real gene expressions and enzyme activities on site. Therefore, the correlation between microbial CUE and carbon quality at a large scale needs further theoretical support. Using stable isotope tracing and indicator species analysis, Buckeridge et al. further evidenced that soil microbial communities may increase CUE by increasing the efficiency of internal compound recycling and microbial necromass recycling, thus contributing to more complex carbon compound accumulation[29]. Additionally, climate may regulate the microbial CUE indirectly by altering forest structure, diversity, productivity, and other properties despite we did not measure these variables. For example, a recent study found a positive association between microbial CUE and tree species diversity in a subtropical forest[30].

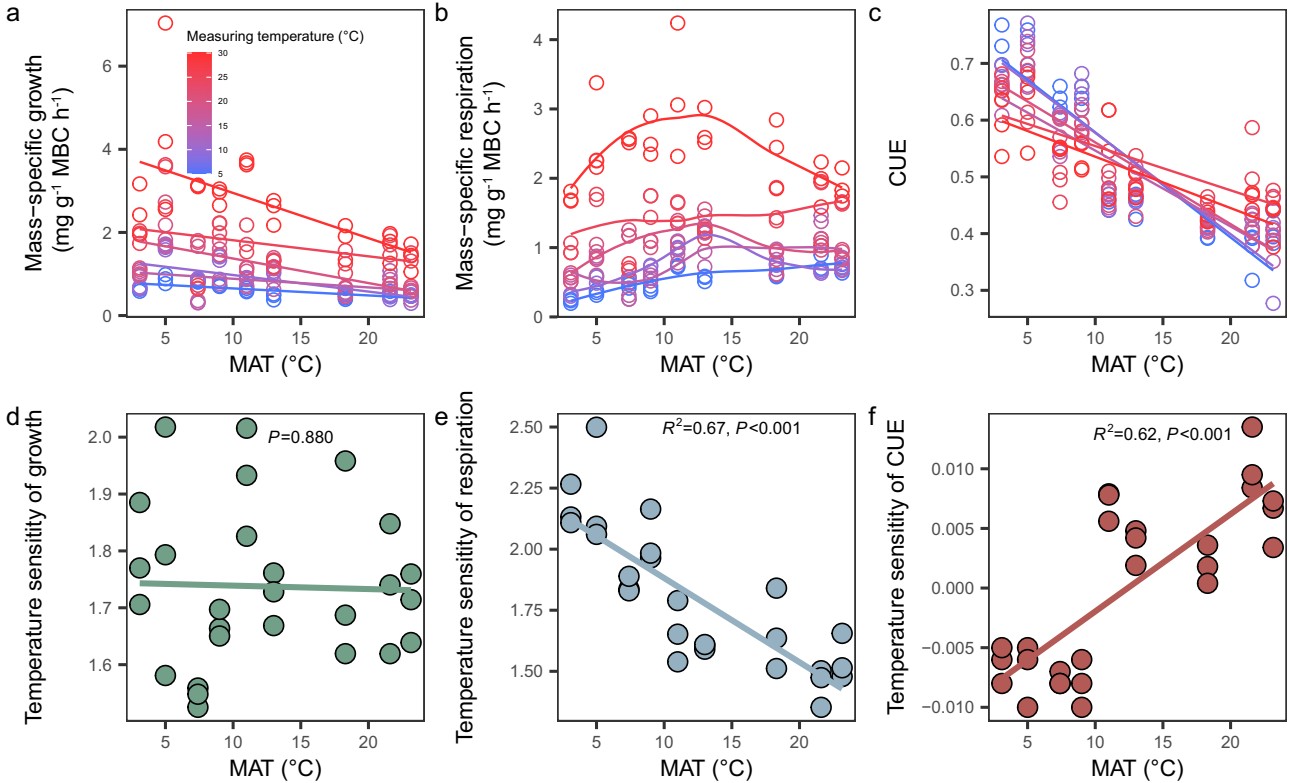

**Fig. 1 | Climate-dependency of microbial physiological traits and their thermal sensitivities. a** Relationship between mass-specific growth (growth per unit microbial biomass carbon) and mean annual temperature (MAT) at different measuring temperatures. **b** Relationship between mass-specific respiration (respiration per unit microbial biomass carbon) and MAT at different measuring temperatures (local polynomial regression). **c** Relationship between microbial carbon use efficiency (CUE) and MAT at different measuring temperatures. **d** Relationship between thermal sensitivity of microbial growth and MAT. **e** Relationship between thermal sensitivity of respiration and MAT. **f** Relationship between thermal sensitivity of microbial CUE and MAT. Relationships are denoted with solid lines and fit statistics ($R^2$ and $P$ values).

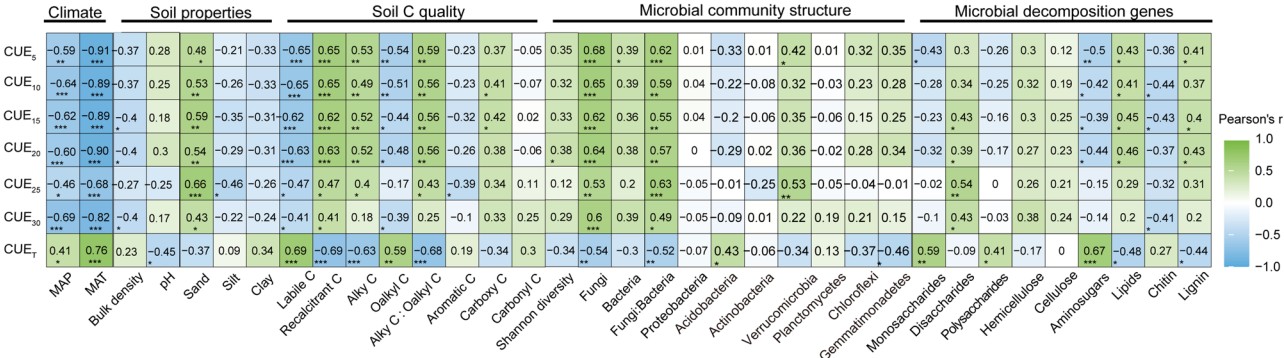

**Fig. 2 | Effects of environmental factors on microbial carbon use efficiency and its thermal sensitivity.** Microbial carbon use efficiency (CUE) and its thermal sensitivity ($CUE_T$) were measured at 5 ($CUE_5$), 10 ($CUE_{10}$), 15 ($CUE_{15}$), 20 ($CUE_{20}$), 25 ($CUE_{25}$), and 30 °C ($CUE_{30}$). MAT, mean annual temperature; MAP, mean annual precipitation. The numbers in the colored squares are the Pearson's coefficients. \*\*\*, $P < 0.001$; \*\*, $P < 0.01$; \*, $P < 0.05$. The specific $P$ values are showed in Supplementary Table 6.

## Climate-dependency of microbial CUE thermal sensitivity

We found that the temperature sensitivity of microbial growth was decoupled from MAT (Fig. 1d). Echoing two previous studies across Chinese forest biomes[31,32], we found a negative correlation between the temperature sensitivity of respiration and MAT (Fig. 1e). Moreover, microbial CUE exhibited a linear response to the measuring temperature (see "Methods"; Supplementary Table 4), aligning with the framework proposed by a previous modeling study[11]. Therefore, the $CUE_T$ is defined as the slope of the linear relationship between microbial CUE and the measuring temperature. Given the patterns in temperature

sensitivities of microbial growth and respiration relative to MAT, we anticipate an increase in microbial $CUE_T$ concurrent with an increase in MAT (Fig. 1f). Accordingly, the temperature optimum for microbial CUE might be lower in colder sites if organisms have adapted to colder climates, whereas the temperature optimum for microbial CUE could be higher in warmer sites if organisms have adapted to warmer climates. Additionally, we frequently observed positive microbial $CUE_T$ in warm forests (Fig. 1f), challenging previous theoretical notions and modeling hypotheses[8,11]. A 27-year-long manipulation experiment at Harvard Forest corroborated our findings, showing that prolonged

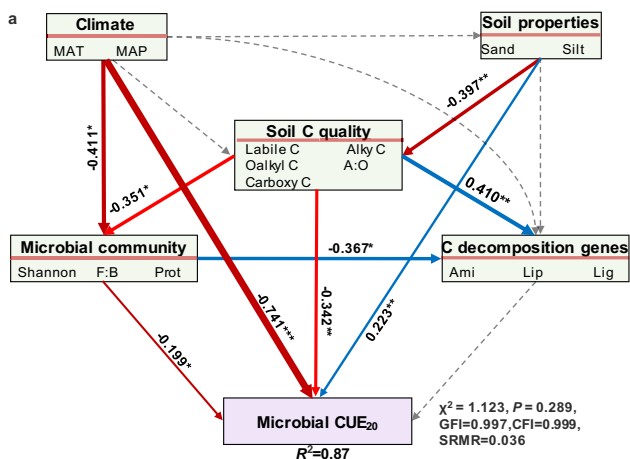
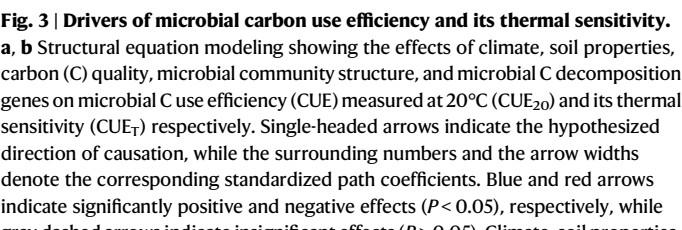

**Fig. 3 | Drivers of microbial carbon use efficiency and its thermal sensitivity.** **a**, **b** Structural equation modeling showing the effects of climate, soil properties, carbon (C) quality, microbial community structure, and microbial C decomposition genes on microbial C use efficiency (CUE) measured at 20°C (CUE$_{20}$) and its thermal sensitivity (CUE$_T$) respectively. Single-headed arrows indicate the hypothesized direction of causation, while the surrounding numbers and the arrow widths denote the corresponding standardized path coefficients. Blue and red arrows indicate significantly positive and negative effects ($P < 0.05$), respectively, while gray dashed arrows indicate insignificant effects ($P > 0.05$). Climate, soil properties,

C quality, microbial community structure, and microbial C decomposition genes are the first components from the principal components analyzes using corresponding factors listed in the rectangles. MAT mean annual temperature; MAP mean annual precipitation; F:B fungi to bacteria ratio; Pro Proteobacteria; Mon Monosaccharides; Ami Aminosugars; Lip Lipids; Lig Lignin; GFI goodness fit index; CFI comparative fit index; SRMR standardized root mean square residual; ***, $P < 0.001$; **, $P < 0.01$; *, $P < 0.05$. The specific $P$ values are showed in Supplementary Table 7 and 8.

warming can shift microbial CUE$_T$ from a negative value to a positive one[4].

Microbial CUE$_T$ has been shown to be species-specific and substrate-specific[2,14,33], thus community-level CUE$_T$ demonstrated here may be undergirded by shifts in community structure, soil carbon quality, as well as the utilization of such carbon sources by microbes, represented by carbon-related functional genes. Both structural equation modeling and Pearson's analysis suggested that soil carbon quality, microbial community structure, and functional genes are essential drivers of microbial CUE$_T$ (Fig. 2 and Fig. 3b). Lower carbon quality is associated with greater temperature sensitivity of respiration, supporting the carbon quality temperature hypothesis[34]. In conjunction with carbon quality independent growth, lower carbon quality resulted in more negative CUE$_T$ (Fig. 2). Consequently, we found that positive microbial CUE$_T$ was associated with functional genes encoding the decomposition of labile components, such as aminosugars, while negative microbial CUE$_T$ was associated with functional genes encoding the decomposition of recalcitrant components, such as lipids and lignin (Fig. 2).

**Implications for soil carbon cycle**
Our forest transect experiment clarified the climate dependency of microbial CUE and its thermal sensitivity. These findings have critical implications for the development of soil carbon models and the SOC feedback to climate warming (Fig. 4). Model structure, parameter value, and initial conditions are the critical factors contributing to the uncertainty across models[35]. Confidence in soil biogeochemical submodels in Earth System Models is low because of uncertainties related to the representation of microbial processes in these models[36]. The linear temperature sensitivity function with a microbial CUE at reference temperature (CUE$_0$) and a temperature response coefficient (CUE$_T$) (i.e., CUE = CUE$_0$ + CUE$_T$ × (T − 20)) has been represented in many carbon models[8,37,38]. However, the parameters of CUE$_0$ and CUE$_T$ in this equation are usually determined from a few experiments by a substrate-dependent approach and are held at constant values[8]. Considering the significant variation of microbial CUE reported in previous studies[28], and the results of stoichiometric modeling[25], even a positive CUE$_0$–MAT relationship, which contrasts with the observation, has

been represented in models[39]. Our study suggests that microbial CUE$_0$ in the previous equation is negatively correlated with MAT, while microbial CUE$_T$ is positively correlated with MAT. Overall, experimental observations of microbial CUE and CUE$_T$ provide an excellent opportunity to evaluate model performance and constrain the uncertainty in model projections. A new generation of microbial-explicit soil carbon models that account for the climate dependency of CUE and CUE$_T$ will likely improve the projections of future soil carbon stocks.

To examine the response of microbial physiology to climate warming, we calculated the SOC-specific carbon uptake (microbial growth plus respiration per unit SOC), SOC-specific growth (microbial growth per unit SOC), and SOC-specific respiration (microbial respiration per unit SOC, reflecting carbon emission/decomposition rate) in both warm (mean annual temperature greater than 10 °C) and cold (mean annual temperature lower than 10 °C) forests (Fig. 4). We found that SOC-specific carbon uptake displayed greater sensitivity to temperature in cold forests than warm forests. Different microbial CUE and its temperature sensitivity in cold and warm forests resulted in differing microbial carbon allocations for growth and respiration. Ultimately, if the temperature is below approximately 24°C, warm forests exhibit a higher carbon emission rate than cold forests at the same temperature. Conversely, if the temperature exceeds approximately 24°C, warm forests have a lower carbon emission rate than cold forests at the same temperature. Overall, a climate-adaptive microbial community appears to have the ability to decrease carbon loss from the soil matrix under corresponding favorable climatic conditions. The plasticity of microbial CUE and its temperature sensitivity modifies the feedback of soil carbon to climate warming.

Our initial broad-scale exploration of microbial CUE$_T$ highlights several future research needs regarding the regulation of microbial CUE for SOC storage. A recent study has suggested a positive contribution from microbial CUE (including microbial CUE calculated by various and incomparable methods) to global SOC storage[7]. Indeed, we identified such positive correlations between microbial CUE and SOC consistently across six measurement temperatures (Supplementary Fig. 6). However, aside from a recent rebuttal based on statistical and process-based model structures considering carbon inputs and

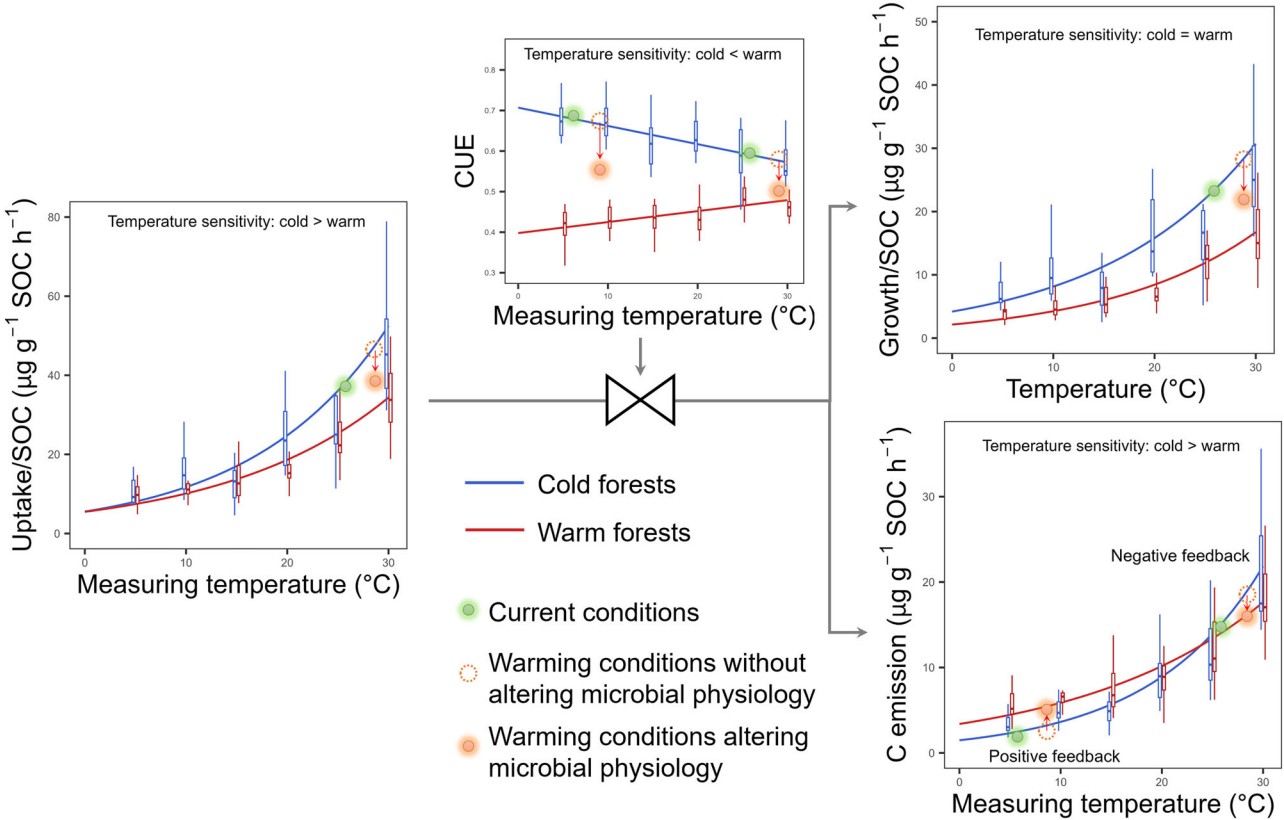

**Fig. 4 | Regulation of microbial carbon use efficiency on soil carbon-climate feedback.** Warm forests had a mean annual temperature greater than 10°C with the positive temperature sensitivity of microbial carbon (C) use efficiency (CUE) (red lines are the fitting values across all warm forests), while cold forests had a mean annual temperature lower than 10°C with the negative temperature sensitivity of microbial CUE (blue lines are the fitting values across all cold forests). Microbial C uptake is the sum of growth and respiration. The red and blue boxes (Centerline, median. Box limits, upper and lower quartiles. Whiskers, 1.5 times interquartile range) are microbial CUE and soil organic C (SOC)-specific C uptake, growth, and respiration in warm and cold forests, respectively. The green dashed, and orange points show the ecological process in current conditions, warming conditions

without altering microbial physiology, and warming conditions altering microbial physiology, respectively, modifying from Singh et al. 2010 (Singh et al., 2010. Nature Reviews Microbiology, 8, 779–790). If there is no adaption of microbial physiology to warming, we could predict the ecological process using the blue line, i.e., the ecological process would shift from the green point (control) to the dashed point (warming). The current study indicates the adaption of microbial physiology, i.e., shifting from green point to orange point. The intersection point of two fitting curves for the relationships between C emission and measuring temperature in warm and cold forests with a measuring temperature of ~24 °C. Source data are provided as a Source Data file.

carbon-quality[40], our findings verify that cold forests with high microbial CUE have a higher carbon emission rate (a more pronounced negative effect on SOC) if the temperature exceeds roughly 24°C (Fig. 4) compared to warmer forests with lower microbial CUE. Thus, the contribution of microbial CUE to SOC may be temperature-dependent. In addition, we should quantify both microbial CUE and $CUE_T$ in a more diverse range of ecosystems beyond the forests of China to provide more precise parameters for microbial models.

## Methods
### Study area and field sampling
The study was conducted in nine forests along a 3425 km north–south transect in China (Supplementary Table 1). The wide-ranging sampling transect led to substantial variations in soil and microbial properties (Supplementary Table 1–2); this provided an ideal natural laboratory for examining soil carbon cycle processes and the mechanisms that underlie them.

Soil samples were collected between July and August 2019. At each site, three sampling plots (50 × 50 m) were established randomly in well-protected national nature reserves to minimize the effect of anthropogenic disturbance. These sites were in areas with relatively homogeneous vegetation, strongly representative of each forest type. Given that topsoil microbes are highly sensitive

to a range of factors associated with climate change, soils from the top 10 cm were collected to study the microbial $CUE_T$ from nine random locations within each plot, after surface litter removal. The nine soil samples were then combined into a composite sample to reduce soil heterogeneity in each plot. After sieving (2 mm diameter), the samples were divided into two subsamples. One subsample was stored at −20°C until the initiation of microbial measurements, while the other subsample was air-dried for chemical and physical analyses.

### Microbial CUE
Soil microbial CUE was assessed using a substrate-independent method predicated on the incorporation of [18]O from water into microbial DNA[24]. Specifically, after a 7-day pre-incubation period in darkness at 25°C[41,42], duplicate aliquots of 500 mg samples of each pre-incubated soil were placed into 2 mL brown chromatographic vials. One vial served as the control for natural [18]O abundance, the other was used for the labeled samples. For one replicate, the [18]O content of soil water was adjusted to 20.0 at% [18]O by adding $H_2^{18}O$; the same volume of unlabeled water was added to another replicate. Subsequently, vials containing the soil samples were placed in 20 mL headspace bottles, with three blank bottles (without soil) per batch of test samples designated as controls. Thereafter, the bottles

were flushed with $CO_2$-free air to achieve headspace $CO_2$ concentrations of approximately 0 ppm. All $^{18}O$ labeled and control samples were then incubated at 60% of their water-holding capacity for 24 h at varying temperatures (5, 10, 15, 20, 25, and 30 °C) to determine the temperature sensitivity of microbial CUE.

After a 24-hour incubation, gas samples were collected from each vial using a syringe. The $CO_2$ concentration was immediately determined using a GC-7890B gas chromatography system (Agilent Technologies). The vials containing labeled and control soils were then capped, immediately frozen, and stored at -80°C until DNA extraction. Total soil DNA was extracted using a FastDNA™ SPIN Kit for Soil, in accordance with the manufacturer's recommendations. Soil DNA concentrations were subsequently determined through the Picogreen fluorescence assay. The remaining DNA extracts were then pipetted into a silver cup and dried at 45°C for five hours to remove all water. The silver capsules were folded and analyzed for $^{18}O$ abundance and total O content using an IRMS-TC/EA (Thermo Scientific) at the Laboratory of Ecological Indicators Analysis (Institute of Geographic Sciences and Nature Resources Research, Chinese Academy of Sciences). Microbial biomass carbon was determined via the $CH_3Cl$ fumigation extraction method, and an extraction efficiency factor of 0.45 was used for calculation.

Microbial growth was estimated by measuring the synthesis of DNA via the incorporation of $^{18}O$ into microbial DNA. Total dsDNA ($DNA_{produced}$, µg) during the 24-h incubation was calculated according to the following equation:

$$DNA_{produced} = O_{Total} \times \frac{at\%_{excess}}{100} \times \frac{100}{at\%_{final}} \times \frac{100}{31.21} \quad (1)$$

where $O_{Total}$ is the total O content (µg) of the dried DNA extract, $at\%_{excess}$ is the at% excess $^{18}O$ of the labeled sample compared with that of the corresponding control. $at\%_{final}$ is the $^{18}O$ at% of soil water at the beginning of incubation (20.0% in our study). The constant 31.21 is the proportional mass of O content of DNA based on the average formula ($C_{39}H_{44}O_{24}N_{15}P_4$). A conversion factor ($f_{DNA}$), calculated as the ratio of soil microbial biomass carbon to DNA content, was used to convert the amount of newly produced DNA ($DNA_{produced}$; µg g$^{-1}$ dry soil) into microbial biomass carbon production after 24 h incubation. Microbial growth ($G$; µg C g$^{-1}$ dry soil h$^{-1}$) was calculated based on the $DNA_{produced}$ and $f_{DNA}$.

$$G = \frac{f_{DNA} \times DNA_{produced}}{DW \times t} \quad (2)$$

where DW (g) is the dry weight of soil and t is the incubation time (h). Moreover, microbial basal respiration rate ($R$, µg C g$^{-1}$ soil h$^{-1}$) was calculated by the following equation:

$$R = \frac{r}{DW \times t} \times \frac{p \times n}{r_c \times T} \times V \quad (3)$$

where $p$ is the atmosphere pressure (kPa), $n$ is the molecular mass of the element C (12.01 g mol$^{-1}$), $r_c$ is the ideal gas constant (8.314 J mol$^{-1}$ K$^{-1}$), and $T$ is the absolute temperature of the gas (295.15 K). $V$ is the headspace volume (L) of the vials. $r$ (ppm) is the amount of $CO_2$ concentration produced during the 24 h incubation period.

Microbial CUE was calculated[18] as follows:

$$CUE = \frac{G}{G + R} \quad (4)$$

## Temperature sensitivity

A commonly used exponential function was adopted to fit changes in microbial respiration and growth with temperature:

$$P_T = a \times e^{cT} \quad (5)$$

$$Q_{10} = e^{10c} \quad (6)$$

where $P_T$ is a specific process at a given temperature, $T$ is temperature in °C, and $a$ and $c$ are model parameters. The parameter "$a$" represents the microbial respiration or growth rate at 0°C, while parameter "$c$" regulates the temperature sensitivity of microbial respiration or growth. To calculate the $CUE_T$, we assessed the relative performance of linear versus the exponential function in accurately depicting the relationship between microbial CUE and assay temperature. We ascertained that the linear function possessed a lower Akaike information criterion compared to the nonlinear function. This implies that microbial CUE demonstrates a linear response to increasing measurement temperature (Supplementary Table 4). Given the mathematical expectations, it is expected that microbial CUE was linear with temperature because both microbial growth and respiration are exponential. Therefore, the $CUE_T$ is the slope of the linear relationship between microbial CUE and the measurement temperature.

## Soil microbial community and functions

The metagenomic sequencing used and any associated references are available in the supplementary online material (Supplementary Text 1). According to results from the KEGG database, the functional annotation and taxonomic assignment from each sample were obtained for further analysis. Information regarding the trends of microbial functional genes in forests across biomes has been shown in our recent study[43]. In it, we defined the functional genes of various carbon complexes, ranging from labile to stable carbon degradation. In general, monosaccharides, disaccharides, polysaccharides, hemicellulose, cellulose, and aminosugar were categorized as having labile carbon composition, while lipids, chitin, and lignin were categorized as having stable carbon composition. However, it remains unclear how the microbial functional genes interacting with carbon quality drive microbial CUE and $CUE_T$. Furthermore, we determined the abundances of soil total bacterial and fungal communities using quantitative PCR of the 16 S rRNA and fungal ITS-1 genes (Supplementary Text 2).

## Soil carbon quality

Data regarding the trends of soil carbon quality were presented in our recent study[43]. Specifically, the chemical structure of soil organic carbon was delineated using solid-state $^{13}C$ cross polarization-magic angle spinning nuclear magnetic resonance spectroscopy via a Bruker 200 Avance spectrometer, outfitted with a 4.7 T wide bore superconducting magnet at a resonance frequency of 50.33 MHz. Spectra were derived using a 3.2 ms 195 w 90 pulse with a contact duration of 1 ms and a recycle delay of 1 s, presets determined based on the T1H value of these samples. Chemical shift regions were defined relative to the methyl resonance of hexamethylbenzene at 17.36 ppm as follows: alkyl carbon (0–45 ppm), O-alkyl carbon (45–110 ppm), aromatic carbon (110–156 ppm), carboxy carbon (156–186 ppm), and carbonyl carbon (186–230 ppm)[44]. Additionally, soil carbon quality was evaluated using an acid hydrolysis method[45]. Specifically, soil samples were subjected to hydrolysis with 20 ml of 2.5 M $H_2SO_4$ at 105°C for 30 min. The hydrolysates were then centrifuged. The residue was rinsed with distilled water and the supernatant was combined with the hydrolysate; this mixture was regarded as labile carbon. The remaining soil residue was rinsed twice with distilled water and dried at 60°C, recorded as the recalcitrant carbon. Higher carbon quality is linked with lower alkyl

carbon to O-alkyl carbon ratio or lower recalcitrant carbon levels (higher labile carbon levels).

## Statistical analyzes

Pearson's correlation analysis was used to investigate the relationships of soil microbial CUE and $CUE_T$ with climates (MAT and mean annual precipitation), soil properties (soil pH, bulk density, and texture), soil carbon quality (labile carbon, recalcitrant carbon, alkyl carbon, O-alkyl carbon, alkyl carbon to O-alkyl carbon ratio, aromatic carbon, carboxy carbon, and carbonyl carbon), microbial community structure (Shannon diversity, fungal abundance, bacterial abundance, fungi to bacteria ratio, and the relative abundances of *Proteobacteria*, *Acidobacteria*, *Actinobacteria*, *Verrucomicrobia*, *Planctomycetes*, *Chloroflexi*, and *Gemmatimonadetes*), and carbon decomposition genes (monosaccharides, disaccharides, polysaccharides, hemicellulose, cellulose, aminosugars, lipids, chitin, lignin).

Then, we used structural equation modeling to further explore the direct and indirect effects of climates, soil properties, soil carbon quality, microbial community structure, and carbon decomposition genes on microbial CUE and its temperature sensitivity. First, an a priori model was proposed, which assumed that the climate factors regulated microbial CUE and its temperature sensitivity directly or indirectly by altering soil properties, soil carbon quality, microbial community structure, and carbon decomposition genes (Supplementary Fig. 7). Second, considering the correlations among factors within each group, we performed principal components analysis within each group to create a new index for climates, soil properties, soil carbon quality, microbial community structure, and carbon decomposition genes[17]. The first component, which accounted for >60% of the variance of each group (the selected variables within each group were shown in Supplementary Table 5), was then introduced in the structural equation modeling[17]. In general, a qualified structural equation modeling is indicated by a non-significant $\chi^2$ test ($P > 0.05$), high goodness fit index ($0.8 < GFI < 1$), high comparative fit index ($CFI > 0.95$), and low standardized root mean square residual ($SRMR < 0.08$)[46,47]. The structural equation modeling was conducted using R packages of *lavaan*.

## Reporting summary

Further information on research design is available in the Nature Portfolio Reporting Summary linked to this article.

# Data availability

All sequences associated with this study are available from the Sequence Read Archive under accession numbers PRJNA977727. The data that supports the findings of this study are openly available in figshare at https://doi.org/10.6084/m9.figshare.25962790.v1. Source data are provided in this paper.

# Code availability

The code that supports the findings of this study is openly available in figshare at https://doi.org/10.6084/m9.figshare.25962697.v1.

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

## Acknowledgements

We thank Jing Wang for her help in $^{18}O$ analysis. This work was financially supported by the National Key Research and Development Program of China (2023YFF1315103), the National Natural Science Foundation of China (No. 42377345). F. Bastida thanks the I + D + I project PID2020-114942RB-I00 funded by MCIN/AEI/10.13039/501100011033 and the i-LINK + 2018 (LINKA20069) from CSIC. M.D-B. acknowledges support from the Spanish Ministry of Science and Innovation for the I + D + i project PID2020-115813RA-I00 funded by MCIN/AEI/10.13039/501100011033. M.D-B. is also supported by a project of the Fondo Europeo de Desarrollo Regional (FEDER) and the Consejería de Transformación Económica, Industria, Conocimiento y Universidades of the Junta de Andalucía (FEDER Andalucía 2014-2020 Objetivo temático "01 - Refuerzo de la investigación, el desarrollo tecnológico y la innovación") associated with the research project P20_00879 (ANDABIOMA).

## Author contributions

All authors contributed intellectual input and assistance to this study. The original concepts were conceived by C.R., Z.Z., G.Y., and G.W. Field management was carried out by C.R., Z.Z., F.Z., and S.Z. Sampling collection and soil characterization were carried out by S.Z., J.W., C.Z., and X.H. Data analyses were done by C.R., Z. Z., J.W., and Y.Y. with the assistance provided by G. Y., G. W., M. D., and F. B. All data analysis and integration were guided by M. D. and F. B. The manuscript was prepared by C.R., Z. Z., G.Y. and G.W.

## Competing interests

The authors declare no competing interests.
