## [Peer Review file · Nature Communications]

REVIEWER COMMENTS

Reviewer #1 (Remarks to the Author):

In this manuscript entitled “negative to positive shifts in thermal sensitivity of soil microbial carbon use efficiency from warm to cold forests”, Ren and co-authors sampled nine sites along a forest transect and used ^{18}O - H_2O method to measure microbial CUE at three temperatures (10, 20 and 30 C). They found microbial CUE increases with decreasing mean annual temperature (MAT), while microbial CUE temperature sensitivity shows an opposite pattern, that is CUE_t increases with increasing MAT. They said those patterns could explain by the difference in soil C quality and microbial C degrading genes as well as fungal to bacterial ratio. While the manuscript was generally prepared well, I have several concerns about it.

First, I am worried about the experimental temperatures for measuring of the CUE. Since we all know the temperature sensitivity of microbial respiration varies with the temperature that assay conducted, the CUE is calculated from respiration and should be also sensitive to the temperature. The authors found a decreasing pattern of CUE along the increasing MAT, which is similar the results from Wang et al. (2021), who did a same study on microbial CUE. However, I noted that Wang et al. (2021) incubated their soils at the temperatures equal to the mean growing season temperature of the site where the sample was collected. That means the CUE was measured under the temperatures which soil microbes have adapting for a long time. Thus, the results from three fixed temperatures (10, 20 and 30 C) in this study can not compare with the results from Wang et al. (2021).

Second, the soil samples were selected from sites with MAT ranging from 3.1 to 23.15 C. But the lowest assay temperature (10 C) was higher than the MAT of four sites (Maor. M, Dongling, Fuxian and Huoditang, in table S1), but it was lower than the MAT of other six sites (from Maoxian to Jianfengling, Table S1). Thus, when calculating the CUE between 20 C and 10 C for each site, it is hard to explain that what is the underlying mechanism for the changes in CUE_t. I do suggest that the real CUE_t of studied sites should be examined under the growing season temperatures and rising a same temperature, for example the Maoer M soils should be incubated at 3.1 and 5.1 C (2 C higher than its MAT) and Jianfengling soils incubated at 23.1 and 25.1, and then to calculate the CUE as well as the CUE_t.

Third, I am not fully understanding how the CUE_t was calculated in the main text (equations 4 and 5). From Lines 255-256, we can see the CUE_t was indicated by the slope of the regression between CUE and assay temperature. But I am not clear that did you calculate the CUE_t for each temperature range, that are CUE₁₀ and CUE₂₀, CUE₂₀ and CUE₃₀, CUE₃₀ and CUE₁₀? Or did you make a regression include the CUE at 10, 20 and 30 C? This should make clear which is very

important for the results of this manuscript. Besides, since the relationships between microbial respiration and temperature or microbial growth and temperature are generally considered as the non-linear (see ref “Predicting the temperature dependence of microbial respiration in soil: A continental-scale analysis” and “Adaptation of soil microbial growth to temperature: Using a tropical elevation gradient to predict future changes”), I think the responses of CUE to increasing temperature may be non-linear. We can see that the CUE in five sites is non-linear increased or decreased with the assay temperatures (Figure 1). Thus, it should be given evidence why selecting linear regression to model the relationship between CUE and temperature.

Fourth, the raw data and code used should be open and shared with the scientific community which could help other to repeat your results.

Line 88, since the ref 15 only includes CUE measured by carbon labelling approaches, it should not be cited and compared with your results.

Line 237-238, the unit of DNA production is ug DNA per g soil dry mass per h. But in the equation 1, there is no parameter of soil mass. In addition, by comparing with Wang et al. (2021), the equation for microbial growth rate has a parameter of soil mass, but it is missed in your equation.

Since I am not confident for the experimental approach and models used to estimate CUEt, I did not provide too many comments for the main text.

Reviewer #2 (Remarks to the Author):

The manuscript “Negative to positive shifts in thermal sensitivity of soil microbial carbon use efficiency from warm to cold forests” provides a look into the patterns and drivers of CUE temperature sensitivity across a latitudinal gradient. I comment the authors on an impressive set of paired measurements of microbial and soil properties, which I think will be of great use and interest to other researchers. There are a few nuances which I think the manuscript would benefit from addressing, however, and which would help with interpretation of the data.

Of particular note:

1. The idea that a more negative CUE temperature sensitivity leads to a positive feedback to climate change comes up a few times in the manuscript (L52-56, 123-126), and is part of a strong focus on CUE as a number rather than a biological phenomenon. In highly simplified models where CUE is as the authors state, just a number partitioning C between soil and the atmosphere without feedback to microbial biomass, then yes there can be a positive feedback to climate if CUE is negative. However, reduced CUE also means less biomass if each cell has a fixed amount of c they will take up or fixed allocation of resources to acquiring that C. So then lower CUE may lead to lower biomass and an overall accumulation of soil C, as modeled in Allison 2014. Therefore, it would be really useful to include mass-specific respiration and growth separately in the supplement and discuss how C goes through the system rather than just focusing on CUE if the authors wish to link their results to climate feedbacks.

2. There are some instances of where the wrong paper has been cited and/or cited as showing the opposite of what the paper actually did show. Please check all references, with particular attention to references 14,15,22,26. Ditto for some of the figures... they do not support the statements made in the manuscript (ex. Fig S7 for L112-113)

3. CUE is unlikely (and not observed here in some cases) to be linear. The temperature optimum for CUE may be lower in cooler sites if organisms are adapted to the temperature. One reason for the more negative CUE in soils from cool places may be that the assay temperatures are higher than the CUE optimum. The potential change in CUE with climate therefore depends on which range the temperature is being shifted.

4. The methods section needs a lot more detail, whether in the main text or supplement. Specific comments are below.

Minor comments:

- Carbon quality is used throughout but needs to be clearly defined and the same definition used throughout. Ex. L 112 aminosugar and lipid degrading genes are defined as recalcitrant, while on L142 it is defined based on absence of labile and O-alkyl C, and L145 it is based on energetic favorability. While some of this information is provided in the methods at the end, it would be useful to explicitly define C quality at the beginning somewhere. And maybe use an existing framework which combines polymerization/need for extracellular enzyme processing with activation energy and energy yield of substrates, such as Gunina and Kuzyakov 2022 (<https://doi.org/10.1111/gcb.16071>)?

- L 97-120: This paragraph is confusing and dense... it brings in the PLS-PM but never really guides the reader through it. It would be helpful to either have an overview guiding the reader through the overall structure and drivers and then after go through individual relationships, or just talk about correlations and not the whole model here, since the model more broadly comes up starting at paragraph 137

- L 105-106: what is "spatial variation of microbial CUE"? Latitudinal gradient?

- L146-147 needs reference

Methods

1. Supplement: what are the SRA numbers for the sequencing data?
2. Supplement: L37-38 If all the metagenomes from all sites were co-assembled and then mapped to, this is problematic as there is no chance the DNA extractions across sites came from the same population.
3. Supplement L 50: Was there an inhibition test completed for qPCR for each DNA extract?
4. Main text L 214: what is “pre-incubation” in this case? At room temperature for 24 hours after removal from the -20C?
5. L232 how was CO₂ quantified?
6. L257: was a cutoff used for the CUET slope fit to determine which slopes were reliable estimates of CUE vs. temperature and which were not?
7. L258-259: needs information on how annotation was completed. MetageneMark just annotates ORFs. How did the authors determine what was a true vs. false annotation?
8. L292: How were the composite variables in the PLS-PM calculated? Were all the variables in figure 2a weighted equally or are there other factors which were removed (ex. MAP)?

Reviewer #3 (Remarks to the Author):

Ren et al. report on relationships between microbial carbon use efficiency (CUE) and CUE temperature sensitivity, climate, soil carbon quality, soil bacterial : fungal ratio, and carbon-degrading genes in soil across a gradient of 9 forest sites. They show that microbial CUE is lower in warmer forests than in colder forests, but that as assay temperature increases microbial CUE actually increases in warm forests (contrary to expectations), while following the expected pattern of decreasing with higher assay temperatures in colder forests. The authors also conduct a thorough analysis of microbial community structure, functional genes, and soil C quality, together with soil property, and use structural equation modelling to interpret these as drivers of microbial CUE and CUET. While I found the methodology overall to be sound, I had some concerns surrounding the motivation and interpretation of the results.

Regarding the motivation: The first paragraph of the introduction makes strong claims about the importance of microbial CUE for future response of soil organic carbon (SOC) stocks to warming. Upon inspection, however, these claims revolve almost entirely around reference 6 (Frey et al.

2013), which does not have empirical evidence to link microbial CUE to soil carbon responses to warming. Data on SOC in that study are only from an unverified modeling exercise, which incorporate and reproduce the authors' assumption of CUE as an important control on SOC stocks under climate change. In fact, while there is a strong theoretical foundation to expect CUE to moderate SOC stocks to warming, interest in describing controls on CUE has far outstripped efforts to link to any actual soil function, including C retention or loss. I would encourage the authors to soften claims in this paragraph and in the discussion and conclusions to focus on a potential or hypothesized role for CUE in moderating SOC stocks, which is not yet supported with evidence, but noting that nevertheless it will likely be important to understand controls on CUE for future efforts to understand C cycle.

Regarding the interpretation: while the discussion offers some linkages of the present findings to previous work, primarily by pointing out when similar patterns have been elsewhere observed, as is revealed in the abstract and the conceptual figure (Figure 3) there is somewhat a lack speculative explanations for the phenomena observed in the data. The finding that microbial CUE declines with higher C quality is in stark contrast to the Cotrufo et al. 2013 opinion paper hypothesizing the opposite, which has arguably driven some of the interest in microbial CUE – yet nothing I could find in the discussion for what could explain this finding. Similarly, the conceptual figure appears to simply repackage the data shown in previous figures so it is all in one place, and not in a format that seems particularly easy to digest. While it is certainly challenging to formulate effective conceptual figures or arrive at an inference about mechanisms, it may be considered by some to be part of the effort of interpreting these kinds of data.

Throughout: The results seem to be reported in present verb tense in the abstract but in past tense elsewhere; suggest to standardize verb tense for reporting results throughout.

Abstract

L29 'We used the [...]' rather than 'We used [...]'

L31 'annual' rather than 'annul'. Please correct this here and throughout, including supplemental information.

Introduction

Overall, the introduction motivates the work (but see caveat about motivations above). However, there is some confusion in using the reference of Frey et al. 2013 to strongly ground and motivate the work with respect to expected temperature-sensitivity of CUE in the first paragraph, then to use

the same study as rationale for weaknesses for assessing the temperature sensitivity of the ^{13}C -CUE method used in the second paragraph. Softening the claims in the first paragraph, and attention to nuance in the second paragraph, may help the authors to overcome this apparent inconsistency.

Also in the second paragraph – referring to the ^{18}O method as ‘novel’ seems uncalled for; the method was published in 2016.

L66 ‘Labile C compounds are’ or ‘Labile C is’

L70 Suggest ‘A large-scale investigation simultaneously considering a wide range of environmental factors, which is currently lacking, would allow [...]’

L74 Suggest ‘Here, we used the ^{18}O - H_2O [...]’

Results

L93 The statement “Here, we found that soil microbial CUE consistently decreased with increasing MAT across different measuring temperatures” appears to contradict the title, which states “Negative to positive shifts in thermal sensitivity of soil microbial carbon use efficiency from warm to cold forests”, which seems to draw more on the patterns shown in CUE in Figure 1 panel (a). Are these broad claims in both the title and the discussion and results able to be reconciled or clarified? Overall, the title seems confusing given how far apart the ‘negative to positive’ and the ‘warm to cold forests’ are from each other, although one seems to refer to the other.

L97: Was a direct effect of MAT on CUE initially included in the model shown in Figure 2? This path is not indicated in the initial diagram shown in the SI (SI Figure 1). Without this path, it does not seem possible to compare MAT effects on CUE as mediated through other measured variables vs. the effect that would emerge in the model as ‘direct’, as the authors appear to do here.

L111-120 As the authors acknowledge, this finding is surprising and appears to contradict a long-held hypothesis about more labile substrates supporting higher microbial CUE. While the authors find evidence for the pattern they observed shown elsewhere (Ref #21), it would also be a more complete intellectual effort here if the authors could offer brief speculation about what mechanisms might explain their observation.

L134-136 Two distinct sentences linked with comma

L137 “Microbial CUET was species-specific and substrate-specific...” reads like a finding the authors report, but since they did only community-level CUE analyses, this statement should be clarified to be about previous literature. “Microbial CUET has been shown to be species-specific and substrate-specific, so community-level CUET shown here may be underlain by shifts in [...]”

L155 This statement isn't clear what is referenced from previous work, or used to explain a finding from the present study; suggest re-phrasing.

L166 “climate-dependency of” rather than “climate-dependent”

Figure 2: In panel a (correlation coefficients), what do relative sizes of boxes represent, and what p-value cutoffs correspond to asterisk for each box? If the legend for correlation coefficients is shifted from the top of panel a to the white space on the lower left of panel c, this would give some space to enlarge the boxes on panel a, which are rather small and difficult to read, and may also allow for the legend to be larger and more legible. How can microbial community structure have a directionality? Please clarify. “Models with different structures were assessed using the Goodness of Fit (GOF) statistic...” This statement seems out of place; both models shown have the same structure.

Figure 3. I found this figure very difficult to interpret, possibly because some indicators seem to be duplicated (the temperature scale) and others unclear (the positive / negative feedback on the side of the soil block). The prevalence of difference C-degrading genes, along with composition of microbial community, is difficult to see because these symbols are all mixed together and overlain on shaded symbol of labile vs. recalcitrant C. If this figure is retained, separating the elements in the soil circles may help them to be more legible.

Methods

L215 ‘mL’ rather than ‘ml’

CUE assays normally carried out at standardized water holding capacities, as water content can strongly affect microbial activity. Was WHC assessed and accounted for in this study? If not, why not?

L227 Please describe what were the 'modified manufacturer's recommendations'. Often 18O-CUE DNA extraction involves extra extraction and retention of all supernatant in the DNA elution steps. Were these steps followed?

L231 Providing methods chronologically can sometimes be helpful. Were not the gas samples taken from the vials before the soils were removed? Suggest to reverse the order in this paragraph. Please provide example of the equation(s) use for gas sample calculations (getting from ppm to mass soil provided by the instrument to mass C respired by soil), as well as the manufacturer of the gas sampling instrument.

L246 superscript for -1

L272 'shown' rather than 'showed'

L285 Missing parentheses; duplicated 'also'

Dear Reviewers:

Thank you very much for considering the manuscript of *Negative to positive shifts in thermal sensitivity of soil microbial carbon use efficiency from warm to cold forests*. We are grateful for the critical comments and suggestions raised by Reviewers on the manuscript, based on which we thoroughly revised it. We had conducted additional measurements of microbial carbon use efficiency at 5, 15, and 25 °C, together with previous measurements (10, 20, and 30 °C), which answered the methodological concerns. In addition, we also improve the discussion and the implication of microbial carbon use efficiency for soil C-climate warming feedbacks. Hope our revision relieved the concerns raised and enhanced the quality of the manuscript. For more details, please refer to point-by-point response to the reviewers' comments (The original Editors and Reviewers' comments are colored blue). In the revised manuscript, the revised parts have been shown in red-inked text.

Reviewer #1 (Remarks to the Author):

General comments

Comment 1: In this manuscript entitled “negative to positive shifts in thermal sensitivity of soil microbial carbon use efficiency from warm to cold forests”, Ren and co-authors sampled nine sites along a forest transect and used 18O-H₂O method to measure microbial CUE at three temperatures (10, 20 and 30 °C). They found microbial CUE increases with decreasing mean annual temperature (MAT), while microbial CUE temperature sensitivity shows an opposite pattern, that is CUE_t increases with increasing MAT. They said those patterns could explain by the difference in soil C quality and microbial C degrading genes as well as fungal to bacterial ratio. While the manuscript was generally prepared well, I have several concerns about it.

First, I am worried about the experimental temperatures for measuring of the CUE. Since we all know the temperature sensitivity of microbial respiration varies with the temperature that assay conducted, the CUE is calculated from respiration and should be also sensitive to the temperature. The authors found a decreasing pattern of CUE along the increasing MAT, which is similar the results from Wang et al. (2021), who did a same study on microbial CUE. However, I noted that Wang et al. (2021) incubated their soils at the temperatures equal to the mean growing season temperature of the site where the sample was collected. That means the CUE was measured under the temperatures which soil microbes have adapting for a long time. Thus, the results from three fixed temperatures (10, 20 and 30 °C) in this study cannot compare with the results from Wang et al. (2021).

Response: We greatly appreciated your positive and constructive comments. Indeed, it should carefully compare our findings and the results from Wang et al. (2021) due to different measuring temperatures. In order to resolve this concern, we had measured microbial CUE at additional temperatures at 5, 15, and 25 °C. The six measuring temperatures (5, 10, 15, 20, 25, and 30 °C) would provide the robust evidence of

latitudinal variation of microbial CUE (**Fig. 1**) (**Line 236-285**). Despite microbial CUE varied with measuring temperatures, cold forests had higher microbial CUE than warm forests, the conclusion of negative correlation between microbial CUE and MAT is maintained (**Fig. 1**).

Fig. 1 Climate-dependency of microbial physiological traits and their thermal sensitivities. **a** Relationship between mass-specific growth (growth per unit microbial biomass) and mean annual temperature (MAT) at different measuring temperatures. **b** Relationship between mass-specific respiration and MAT at different measuring temperatures (local polynomial regression). **c** Relationship between microbial C use efficiency (CUE) and MAT at different measuring temperatures. **d** Relationship between thermal sensitivity of microbial growth and MAT. **e** Relationship between thermal sensitivity of respiration and MAT. **f** Relationship between thermal sensitivity of microbial CUE and MAT.

Comment 2: Second, the soil samples were selected from sites with MAT ranging from 3.1 to 23.15 C. But the lowest assay temperature (10 C) was higher than the MAT of four sites (Maor. M, Dongling, Fuxian and Huoditang, in table S1), but it was lower than the MAT of other six sites (from Maoxian to Jianfengling, Table S1). Thus, when calculating the CUE between 20 C and 10 C for each site, it is hard to explain that what is the underlying mechanism for the changes in CUE. I do suggest that the real CUE of studied sites should be examined under the growing season temperatures and rising a same temperature, for example the Maor M soils should be incubated at 3.1 and 5.1 C (2 C higher than its MAT) and Jianfengling soils incubated at 23.1 and 25.1, and then to calculate the CUE as well as the CUEt

Response: Thanks for your suggestions. Here, using six measuring temperatures (5, 10, 15, 20, 25, and 30 °C), we found that the relationship between microbial CUE and measuring temperature is linear (please refer to our **response to your 3rd comments**), therefore, the current experimental design can reveal the temperature sensitivity of

microbial CUE. In addition, the slope of linear relationships between microbial CUE and temperature is small. It is difficult to accurately simulate the effect of in situ warming (such as 2°C higher than its respective MAT) on microbial CUE considering the small slope and random errors.

Comment 3: Third, I am not fully understanding how the CUE_T was calculated in the main text (equations 4 and 5). From Lines 255-256, we can see the CUE_T was indicated by the slope of the regression between CUE and assay temperature. But I am not clear that did you calculate the CUE_T for each temperature range, that are CUE₁₀ and CUE₂₀, CUE₂₀ and CUE₃₀, CUE₃₀ and CUE₁₀? Or did you make a regression include the CUE at 10, 20 and 30 C? This should make clear which is very important for the results of this manuscript. Besides, since the relationships between microbial respiration and temperature or microbial growth and temperature are generally considered as the non-linear (see ref “Predicting the temperature dependence of microbial respiration in soil: A continental-scale analysis” and “Adaptation of soil microbial growth to temperature: Using a tropical elevation gradient to predict future changes”), I think the responses of CUE to increasing temperature may be non-linear. We can see that the CUE in five sites is non-linear increased or decreased with the assay temperatures (Figure 1). Thus, it should be given evidence why selecting linear regression to model the relationship between CUE and temperature.

Response: Thanks for your suggestions. It is a critical question to explore the non-linear vs. linear relationships between microbial CUE and measuring temperature. Our original measurement of microbial CUE at three temperatures cannot robustly answer this question. Therefore, we added three additional measuring measurements at 5, 15, and 25 °C. We compared the performances of linear function and non-function (exponential function) based on Akaike information criterion (AIC). We found that linear function had the lower AIC than non-linear function, i.e., microbial CUE linearly response to increasing measuring temperature (Supplementary Table 3). Mathematically, it is also expected that microbial CUE is linearly increased with temperature because both microbial growth and respiration are exponential related to temperature. In addition, we had revised the method section accordingly (**Line 293-299**) and provided the original data accompanied by the revised manuscript.

Supplementary Table 3 Akaike information criterion (AIC) estimates for linear and non-linear models of microbial CUE_T

Study sites	Linear function	Non-linear function
Maoer M.	-65.2473	-60.7426
Dongling M.	-58.0453	-50.7939
Fuxian	-62.8806	-50.0061
Huoditang	-76.3749	-63.2925
Maoxian	-57.5239	-53.5315
Gongga M.	-78.6382	-72.5459
Ailao M.	-92.6791	-85.8054
Xishuangbanna	-58.148	-47.9978
Jianfengling	-57.0253	-53.7097

Comment 4: Fourth, the raw data and code used should be open and shared with the scientific community which could help other to repeat your results.

Response: Thanks. We have uploaded our data with the revised manuscript.

Specific comments:

Comment 5: Line 88, since the ref 15 only includes CUE measured by carbon labelling approaches, it should not be cited and compared with your results.

Response: Thanks, revised.

Comment 6: Line 237-238, the unit of DNA production is ug DNA per g soil dry mass per h. But in the equation 1, there is no parameter of soil mass. In addition, by comparing with Wang et al. (2021), the equation for microbial growth rate has a parameter of soil mass, but it is missed in your equation

Response: We have added in the main text (**Line 263-275**).

Comment 7: Since I am not confident for the experimental approach and models used to estimate CUEt, I did not provide too many comments for the main text.

Response: Thanks for your critical comments on our experimental approach and calculation of temperature sensitivity of microbial CUE, which significantly increased the quality of our study. We also improved the organization and writing of the manuscript according to your and other two reviewers' suggestions.

Reviewer #2 (Remarks to the Author):

General comments

Comment 1: The manuscript “Negative to positive shifts in thermal sensitivity of soil microbial carbon use efficiency from warm to cold forests” provides a look into the patterns and drivers of CUE temperature sensitivity across a latitudinal gradient. I comment the authors on an impressive set of paired measurements of microbial and soil properties, which I think will be of great use and interest to other researchers. There are a few nuances which I think the manuscript would benefit from addressing, however, and which would help with interpretation of the data

The idea that a more negative CUE temperature sensitivity leads to positive feedback to climate change comes up a few times in the manuscript (L52-56, 123-126), and is part of a strong focus on CUE as a number rather than a biological phenomenon. In highly simplified models where CUE is as the authors state, just a number partitioning C between soil and the atmosphere without feedback to microbial biomass, then yes there can be positive feedback to climate if CUEt is negative. However, reduced CUE also means less biomass if each cell has a fixed amount of c they will take up or fixed allocation of resources to acquiring that C. So then lower CUE may lead to lower biomass and an overall accumulation of soil C, as modeled in Allison 2014. Therefore, it would be useful to include mass-specific respiration and growth separately in the supplement and discuss how C goes through the system rather than just focusing on CUE if the authors wish to link their results to climate feedbacks.

Response: Thanks for your positive and insight suggestions. First, according to your constructive suggestions, we analyzed the mass-specific growth (growth per unit microbial biomass), mass-specific respiration (respiration per unit microbial biomass), and temperature sensitivities of respiration and growth. We found that soil mass-specific microbial growth (microbial growth per unit microbial biomass) was decreased as increasing MAT despite the measuring temperatures (**Fig. 1a**). This pattern is agreement with the latitudinal compensation hypothesis from macro-ecology, i.e., organisms from cold environments enhance their basal metabolic rates and have high potential growth rates to compensate for the short period of growing seasons (Levinton, 1983; Conover & Present, 1990; Yamahira et al., 2007). Contrasting with microbial growth, trends of mass-specific respiration along MAT varied with measuring temperatures (**Fig. 1b**). Temperature sensitivity of growth was decoupled with MAT (**Fig. 1d**), while temperature sensitivity of respiration was decreased as increasing MAT (**Fig. 1e**). Consequently, we found the positive correlation between temperature sensitivity of microbial CUE and MAT (**Fig. 1f**). More detail please refer to our **responses to Reviewer #1 1st comment**.

Second, in order to clearly answer the feedback of microbial physiology to climate warming, we calculated the SOC-specific C assimilation (growth plus respiration), growth, and respiration (reflects C emission/decomposition rate) (**Fig. 4**). We found that SOC-specific C assimilation was more sensitive to temperature in cold forests than warm forests, resulting higher SOC-specific C assimilation at same temperature in the former forests than the latter forests. However, different microbial CUE and its temperature sensitivity results in different microbial C allocations in growth and respiration. If temperature is lower than $\sim 24^{\circ}\text{C}$, warm forests had greater C emission rate (SOC-specific respiration) than cold forests at same temperature. However, if temperature is greater than $\sim 24^{\circ}\text{C}$, warm forests had lower C emission rate than cold forests at same temperature. Overall, climate-adaptive microbial community likely has a capacity to reduce C loss from soil matrix under corresponding “comfortable” climate conditions, plasticity of microbial CUE and its temperature sensitivity alter the feedback of soil C to climate warming.

Fig. 4 Regulation of microbial C use efficiency on soil C-climate feedbacks. Warm forests had mean annual temperature greater than 10°C with positive temperature sensitivity of microbial C use efficiency (red lines are the fitting values across all warm forests), while cold forests had mean annual temperature lower than 10°C with negative temperature sensitivity of microbial CUE (blue lines are the fitting values across all cold forests). Microbial C assimilation is the sum of growth and respiration.

References:

- Levinton, J. S. The latitudinal compensation hypothesis—growth data and a model of latitudinal growth-differentiation based upon energy budgets. 1. interspecific comparison of ophryotrocha (polychaeta, dorvilleidae). *Biol. Bull.* 165, 686–698 (1983).
- Conover, D. O. & Present, T. M. C. Countergradient variation in growth-rate—compensation for length of the growing-season among atlantic silversides from different latitudes. *Oecologia* 83, 316–324 (1990).
- Yamahira, K., Kawajiri, M., Takeshi, K. & Irie, T. Inter- and intrapopulation variation in thermal reaction norms for growth rate: Evolution of latitudinal compensation in ectotherms with a genetic constraint. *Evolution* 61, 1577–1589 (2007).

Comment 2: There are some instances of where the wrong paper has been cited and/or cited as showing the opposite of what the paper actually did show. Please check all references, with particular attention to references 14,15,22,26. Ditto for some of the figures... they do not support the statements made in the manuscript (ex. Fig S7 for L112-113)

Response: Thanks, we checked the citation throughout the revised manuscript.

Comment 3: CUEt is unlikely (and not observed here in some cases) to be linear. The temperature optimum for CUE may be lower in cooler sites if organisms are adapted to the temperature. One reason for the more negative CUEt in soils from cool places may be that the assay temperatures are higher than the CUE optimum. The potential change in CUE with climate therefore depends on which range the temperature is being shifted

Response: Thanks for your constructive suggestions. We had added another three measuring temperatures at 5, 15, and 25 °C. Using six measuring temperatures (5, 10, 15, 20, 25, and 30 °C), we evidenced that microbial CUE is linear correlated with measuring temperature. More detail please refer to our **responses to Reviewer #1 3rd comment**.

Specific comments:

Comment 4: Carbon quality is used throughout but needs to be clearly defined and the same definition used throughout. Ex. L 112 aminosugar and lipid degrading genes are defined as recalcitrant, while on L142 it is defined based on absence of labile and O-alkyl C, and L145 it is based on energetic favorability. While some of this information is provided in the methods at the end, it would be useful to explicitly define C quality at the beginning somewhere. And maybe use an existing framework which combines polymerization/need for extracellular enzyme processing with activation energy and energy yield of substrates, such as Gunina and Kuzyakov 2022 (<https://doi.org/10.1111/gcb.16071>)?

Response: Thanks for your suggestions. First, we defined the carbon quality as early as possible (**Line 88-91**). Second, it is an important paper for our understanding of microbial regulation of soil C cycling from Gunina and Kuzyakov, 2022. We calculated the oxidation state of soil C from the solid-state ¹³C cross polarization magic angle spinning (CP/MAS) NMR spectroscopy. However, we found that both microbial CUE and its temperature sensitivity were decoupled with the oxidation state of soil C. Therefore, we did not include such analysis in the revised manuscript (**showed in the following figure**).

Comment 5: L97-120: This paragraph is confusing and dense... it brings in the PLS-PM but never really guides the reader through it. It would be helpful to either have an overview guiding the reader through the overall structure and drivers and then after go through individual relationships, or just talk about correlations and not the whole model here, since the model more broadly comes up starting at paragraph 137

Response: Thanks for your suggestions, revised (Line 112-128).

Comment 6: what is “spatial variation of microbial CUE”? Latitudinal gradient?

Response: Thanks for your suggestions. Revised (Line 100).

Comment 7: needs reference

Response: Thanks for your suggestions. Revised.

Comment 8: Supplement: what are the SRA numbers for the sequencing data?

Response: Thanks for your suggestions. We had uploaded the sequencing data and provided the SRA number (PRJNA977727) (Line 498).

Comment 9: Supplement: L37-38 If all the metagenomes from all sites were co-assembled and then mapped to, this is problematic as there is no chance the DNA extractions across sites came from the same population.

Response: We are sorry for this confusing expression. Not all sample data is mixed and sequenced, instead, each sample is individually assembled first, then mixing the unused reads from the single spell together to obtain more contigs data. The clean data and assembly status were shown in the **following table**, which have been showed in our recent study (Ren et al., 2022).

References:

Ren, C. J. et al. Microbial traits determine soil C emission in response to fresh carbon inputs in forests across biomes. *Global Change Biology* 28, 1516-1528 (2022)

Forest site	Raw reads	HQ Reads (%)	GC (%)	Q20(%)	Q30 (%)	Contigs				Scaffolds			
						Total (bp)	Max(bp)	Min(bp)	N50 (bp)	Total (bp)	Max(bp)	Min (bp)	N50 (bp)
ME1	80054432	99.92	63.45	98.09	94.51	1581540	50299	200	434	1566969	50299	200	436
ME2	81215534	99.90	63.26	97.82	93.95	1533720	11858	200	442	1517774	13304	200	445
ME3	84255180	99.94	63.87	98.36	95.26	2066167	19298	200	425	2059476	19298	200	426
DL1	80542294	99.88	63.94	97.86	94.08	1566292	9623	200	426	1554350	9816	200	428
DL2	82246566	99.89	63.85	97.91	94.18	1772577	300578	200	449	1750173	526211	200	453
DL3	81535110	99.90	64.36	97.86	94.07	1498486	10853	200	435	1483699	12556	200	437
FX1	91266842	99.88	64.54	97.86	94.08	2465349	61327	200	478	2423358	61834	200	484
FX2	85343740	99.89	64.18	97.94	94.27	2290786	372215	200	493	2251225	643370	200	500
FX3	85507122	99.88	64.34	98.06	94.61	2342118	142385	200	495	2302463	190396	200	502
HDT1	72304392	99.88	63.24	97.84	93.96	1129631	42100	200	423	1117548	42100	200	426
HDT2	78006954	99.90	62.44	97.70	93.63	1421478	20797	200	436	1404296	35023	200	439
HDT3	79620646	99.89	60.97	97.92	94.14	1971089	69135	200	536	1933460	71137	200	546
MX1	83553110	99.91	62.65	98.04	94.38	2327517	344886	200	605	2257302	386637	200	627
MX2	74453044	99.89	61.60	97.77	93.80	1910211	208669	200	550	1855434	323465	200	566
MX3	82683192	99.94	62.37	98.26	94.96	2629169	112187	200	495	2601537	112187	200	499

GG1	85095326	99.86	64.85	97.80	93.98	1596432	21731	200	421	1584710	21731	200	423
GG2	81275762	99.88	63.47	97.88	94.09	1300387	37744	200	419	1289752	37744	200	421
GG3	78531536	99.88	63.54	97.80	93.93	1461172	46859	200	437	1444090	65355	200	440
AL1	88638876	99.87	63.67	97.88	94.14	2299659	129569	200	612	2232228	151440	200	633
AL2	91537620	99.87	63.79	97.87	94.09	2421494	56896	200	523	2363824	56896	200	533
AL3	90368240	99.88	63.22	98.02	94.48	2047837	162663	200	545	2000868	162663	200	556
XSBN1	77379530	99.90	60.75	97.98	94.27	1906871	42128	200	546	1864157	51442	200	557
XSBN2	84879412	99.91	61.33	97.79	93.84	2211066	48533	200	524	2165393	61975	200	534
XSBN3	85335556	99.90	61.08	97.90	94.09	2037739	62457	200	524	1996562	62457	200	533
JFL1	95397404	99.89	63.72	97.95	94.33	2019770	99981	200	458	1989411	99981	200	463
JFL2	90928246	99.90	64.16	98.09	94.63	2339264	69147	200	476	2309970	69147	200	480
JFL3	89968786	99.90	64.16	97.98	94.38	1989668	48865	200	456	1960986	81992	200	461

Comment 10: Supplement L 50: Was there an inhibition test completed for qPCR for each DNA extract?

Response: Yes, each DNA extract was completed for qPCR.

Comment 11: Main text L 214: what is “pre-incubation” in this case? At room temperature for 24 hours after removal from the -20C?

Response: We are sorry for this incomplete expression. In our study, before ^{18}O -H₂O tracer, we conducted 7-day pre-incubation in darkness at 25°C, aiming to revive microbial metabolism at low temperatures (stored at -20 °C) (Morrissey et al, 2017, Feng et al, 2022). Afterwards, all ^{18}O labeled and control samples were incubated for 24 h at the temperature of 5, 10, 15, 20, 25 or 30°C to determine the temperature sensitivity of microbial CUE (**Line 236-238, Line 246-248**).

References:

- Morrissey, E. M. et al. Bacterial carbon use plasticity, phylogenetic diversity, and the priming of soil organic matter. *ISME. J* 11, 1890-1899 (2017).
- Feng, X. H. et al. Nitrogen input enhances microbial carbon use efficiency by altering plant-microbe-mineral interactions. *Glob. Change. Biol* 28, 4845-4860 (2022).

Comment 12: L232 how was CO₂ quantified?

Response: In our study, the CO₂ concentration was determined immediately using a gas chromatograph system (GC-7890B; Agilent Technologies). We also added the detail of the calculation of respiration in the revised manuscript (**L254-256**).

Comment 13: L257: was a cutoff used for the CUEt slope fit to determine which slopes were reliable estimates of CUE vs. temperature and which were not?

Response: Thanks for your suggestions. Please refer to our **responses to Reviewer #1 3rd comment**.

Comment 14: L258-259: needs information on how annotation was completed. MetageneMark just annotates ORFs. How did the authors determine what was a true vs. false annotation?

Response: We have added the detail of annotation as followed (**Line 302-309**):

According to results from the KEGG database, the functional annotation and taxonomic assignment from each sample were obtained for further analysis. Some information regarding the trends of microbial functional genes in forests across biomes have been showed in our recent study (Ren et al., 2022), where the functional genes of various C-complexes ranging from labile to stable C degradation were defined, in general, monosaccharides, disaccharides, polysaccharides, hemicellulose, cellulose, and aminosugar were categorized as having labile C composition, while lipids, chitin, and lignin were categorized as having stable C composition.

References:

Ren, C. J. et al. Microbial traits determine soil C emission in response to fresh carbon inputs in forests across biomes. *Global Change Biology* 28, 1516-1528 (2022)

Comment 15: L292: How were the composite variables in the PLS-PM calculated? Were all the variables in figure 2a weighted equally or are there other factors which were removed (ex. MAP)?

Response: Thanks for your constructive suggestions, we had revised the analysis associated with structural equation modeling (**Line 340-354**).

Reviewer #3 (Remarks to the Author):

Comment 1: Ren et al. report on relationships between microbial carbon use efficiency (CUE) and CUE temperature sensitivity, climate, soil carbon quality, soil bacterial:fungal ratio, and carbon-degrading genes in soil across a gradient of 9 forest sites. They show that microbial CUE is lower in warmer forests than in colder forests, but that as assay temperature increases microbial CUE actually increases in warm forests (contrary to expectations), while following the expected pattern of decreasing with higher assay temperatures in colder forests. The authors also conduct a thorough analysis of microbial community structure, functional genes, and soil C quality, together with soil property, and use structural equation modelling to interpret these as drivers of microbial CUE and CUET. While I found the methodology overall to be sound, I had some concerns surrounding the motivation and interpretation of the results.

Regarding the motivation: The first paragraph of the introduction makes strong claims about the importance of microbial CUE for future response of soil organic carbon (SOC) stocks to warming. Upon inspection, however, these claims revolve almost entirely around reference 6 (Frey et al. 2013), which does not have empirical evidence to link microbial CUE to soil carbon responses to warming. Data on SOC in that study are only from an unverified modeling exercise, which incorporate and reproduce the authors' assumption of CUE as an important control on SOC stocks under climate change. In fact, while there is a strong theoretical foundation to expect CUE to moderate SOC stocks to warming, interest in describing controls on CUE has far outstripped efforts to link to any actual soil function, including C retention or loss. I would encourage the authors to soften claims in this paragraph and in the discussion and conclusions to focus on a potential or hypothesized role for CUE in moderating SOC stocks, which is not yet supported with evidence, but noting that nevertheless it

will likely be important to understand controls on CUE for future efforts to understand C cycle.

Response: Thanks for your support of our manuscript and insight suggestions. First, we provide the balanced view of the regulation of microbial CUE on soil C (positive vs negative) according to your and **Reviewer #2' comments**. Second, we did our best to answer the potential role for microbial CUE in moderating soil C cycling using our experimental data, more detail please refer to our **response to Reviewer #2' 1st comment**.

Comment 2: Regarding the interpretation: while the discussion offers some linkages of the present findings to previous work, primarily by pointing out when similar patterns have been elsewhere observed, as is revealed in the abstract and the conceptual figure (Figure 3) there is somewhat a lack speculative explanations for the phenomena observed in the data. The finding that microbial CUE declines with higher C quality is in stark contrast to the Cotrufo et al. 2013 opinion paper hypothesizing the opposite, which has arguably driven some of the interest in microbial CUE – yet nothing I could find in the discussion for what could explain this finding. Similarly, the conceptual figure appears to simply repackage the data shown in previous figures so it is all in one place, and not in a format that seems particularly easy to digest. While it is certainly challenging to formulate effective conceptual figures or arrive at an inference about mechanisms, it may be considered by some to be part of the effort of interpreting these kinds of data

Response: Thanks for your critical comments. We had substantially improved the discussion of the manuscript. First, we did find that soils in cold forests with low C quality had high microbial CUE, which is contrast to the Microbial Efficiency-Matrix Stabilization (Cotrufo et al., 2013). We also calculated the oxidation state of soil C according to **Reviewer #2' 4th comment**, and found that both microbial CUE and its temperature sensitivity were decoupled with the oxidation state of soil C. Here, we suggest that the latitudinal gradient of microbial CUE is predominantly explained by the MAT directly. In addition, microbial community structure and soil properties also explain the variation of microbial CUE. However, we found that C-quality explain the latitudinal gradient of temperature sensitivity of microbial CUE, because soils in cold forests with low C quality had high temperature sensitivity of respiration, supporting the supporting the C quality temperature hypothesis (Davidson et al., 2006) (**Line 165-167**).

Second, we also improved the implications of our findings for soil C-climate change feedbacks and replaced the original conceptual figure, more detail please refer to our **response to Reviewer #2' 1st comment**.

References:

Cotrufo, M. F., Wallenstein, M. D., Boot, C. M., Deneff, K. & Paul, E. The Microbial Efficiency-Matrix Stabilization (MEMS) framework integrates plant litter decomposition with soil organic matter stabilization: do labile plant inputs form stable soil organic matter? *Glob. Change. Biol* 19, 988-995 (2013).

Davidson, E. A. & Janssens, I. A. Temperature sensitivity of soil carbon decomposition and feedbacks to climate change. *Nature* 440, 165-173 (2006).

Comment 3: Throughout: The results seem to be reported in present verb tense in the abstract but in past tense elsewhere; suggest to standardize verb tense for reporting results throughout

Response: Thanks, we had standardized the verb tense throughout the manuscript.

Specific comments:

Comment 4: L29 ‘We used the [...]’ rather than ‘We used [...]’

Response: Thanks, revised.

Comment 5: L31 ‘annual’ rather than ‘annul’. Please correct this here and throughout, including supplemental information

Response: Thanks, revised.

Comment 6: Introduction: Overall, the introduction motivates the work (but see caveat about motivations above). However, there is some confusion in using the reference of Frey et al. 2013 to strongly ground and motivate the work with respect to expected temperature-sensitivity of CUE in the first paragraph, then to use the same study as rationale for weaknesses for assessing the temperature sensitivity of the ¹³C-CUE method used in the second paragraph. Softening the claims in the first paragraph, and attention to nuance in the second paragraph, may help the authors to overcome this apparent inconsistency

Response: Thanks for your insight suggestion, we hand improved the introduction. Overall, the first paragraph focuses on the debatable regulation of microbial CUE on soil C storage and the lack of spatial variation of microbial CUE considering the limitations of method. In the second paragraph, we focused on the motivates of temperature sensitivity of microbial CUE and its potential role in soil C-climate warming feedbacks.

Comment 7: Also in the second paragraph – referring to the ¹⁸O method as ‘novel’ seems uncalled for; the method was published in 2016

Response: Thanks, we have deleted the ‘novel’.

Comment 8: L66 ‘Labile C compounds are’ or ‘Labile C is’

Response: Thanks, revised.

Comment 9: L74 Suggest ‘Here, we used the ¹⁸O-H₂O [...]’

Response: Thanks, revised.

Comment 10: L93 The statement “Here, we found that soil microbial CUE consistently decreased with increasing MAT across different measuring temperatures” appears to contradict the title, which states “Negative to positive shifts in thermal sensitivity of

soil microbial carbon use efficiency from warm to cold forests”, which seems to draw more on the patterns shown in CUE in Figure 1 panel (a). Are these broad claims in both the title and the discussion and results able to be reconciled or clarified? Overall, the title seems confusing given how far apart the ‘negative to positive’ and the ‘warm to cold forests’ are from each other, although one seems to refer to the other

Response: Thanks for your suggestions. First, the title was revised as “Thermal sensitivity of soil microbial carbon use efficiency across forest biomes”. Second, we revised the corresponding expression throughout the revised manuscript.

Comment 11: L97: Was a direct effect of MAT on CUE initially included in the model shown in Figure 2? This path is not indicated in the initial diagram shown in the SI (SI Figure 1). Without this path, it does not seem possible to compare MAT effects on CUE as mediated through other measured variables vs. the effect that would emerge in the model as ‘direct’, as the authors appear to do here

Response: Thanks for your suggestions. We have added the direct effect of MAT on microbial CUE and its temperature sensitivity. And we found that MAT significantly regulate microbial CUE and its temperature sensitivity through direct and indirect pathways (**Fig. 3; Supplementary Fig. 2**).

Comment 12: L111-120 As the authors acknowledge, this finding is surprising and appears to contradict a long-held hypothesis about more labile substrates supporting higher microbial CUE. While the authors find evidence for the pattern they observed shown elsewhere (Ref #21), it would also be a more complete intellectual effort here if the authors could offer brief speculation about what mechanisms might explain their observation

Response: Thanks for your suggestions, please refer to our response to **your 2nd comment**.

Comment 13: L134-136 Two distinct sentences linked with comma

Response: Thanks, revised.

Comment 14: L137 “Microbial CUET was species-specific and substrate-specific...” reads like a finding the authors report, but since they did only community-level CUE analyses, this statement should be clarified to be about previous literature. “Microbial CUET has been shown to be species-specific and substrate-specific, so community-level CUET shown here may be underlain by shifts in [...]”

Response: Thanks, revised.

Comment 15: L155 This statement isn’t clear what is referenced from previous work, or used to explain a finding from the present study; suggest re-phrasing

Response: Sorry for this comment, this comment you raised is not very consistent with the corresponding line in the original text.

Comment 16: L166 “climate-dependency of” rather than “climate-dependent”

Response: Thanks, revised.

Comment 17: Figure 2: In panel a (correlation coefficients), what do relative sizes of boxes represent, and what p-value cutoffs correspond to asterisk for each box? If the legend for correlation coefficients is shifted from the top of panel a to the white space on the lower left of panel c, this would give some space to enlarge the boxes on panel a, which are rather small and difficult to read, and may also allow for the legend to be larger and more legible. How can microbial community structure have a directionality? Please clarify. “Models with different structures were assessed using the Goodness of Fit (GOF) statistic...” This statement seems out of place; both models shown have the same structure

Response: Thanks. First, we had revised the correlation plot (**Fig. 2**). Second, in the revised manuscript, microbial community structure in the finally SEM includes Shannon diversity, fungi:bacteria, and the relative abundance of *Proteobacteria*. We also provided the direction of individual factors within each group (**Fig. 3**). Third, we have changed the expression and method of SEM, and re-constructed SEM to identify the direct and indirect effects of environmental predictors on microbial CUE and CUE_T. Finally, a qualified SEM modeling is indicated by a non-significant χ^2 test ($P > 0.05$), high goodness fit index ($0.8 < GFI < 1$), high comparative fit index ($CFI > 0.95$), and low standardized root mean square residual ($SRMR < 0.08$).

Comment 18: Figure 3. I found this figure very difficult to interpret, possibly because some indicators seem to be duplicated (the temperature scale) and others unclear (the positive / negative feedback on the side of the soil block). The prevalence of difference C-degrading genes, along with composition of microbial community, is difficult to see because these symbols are all mixed together and overlain on shaded symbol of labile vs. recalcitrant C. If this figure is retained, separating the elements in the soil circles may help them to be more legible

Response: Thanks for your suggestions. We had revised this figure according to you and **Reviewer #2'** comments. More detail please refer to our response to **Reviewer #2' 1st comment**.

Comment 19: L215 'mL' rather than 'ml'

Response: Thanks, revised.

Comment 20: CUE assays normally carried out at standardized water holding capacities, as water content can strongly affect microbial activity. Was WHC assessed and accounted for in this study? If not, why not?

Response: Thank you for your suggestion. All ¹⁸O-H₂O labeled and control samples were incubated under 60% water holding capacity. Revised (**Line 247**).

Comment 21: L227 Please describe what were the 'modified manufacturer's recommendations. Often ¹⁸O-CUE DNA extraction involves extra extraction and retention of all supernatant in the DNA elution steps. Were these steps followed?

Response: We are sorry for this inappropriate expression; total soil DNA was extracted with a DNA extraction kit (FastDNA™ SPIN Kit for Soil) following the manufacturer's re-recommendations. Revised.

Comment 22: L231 Providing methods chronologically can sometimes be helpful. Were not the gas samples taken from the vials before the soils were removed? Suggest to reverse the order in this paragraph. Please provide example of the equation(s) use for gas sample calculations (getting from ppm to mass soil provided by the instrument to mass C respired by soil), as well as the manufacturer of the gas sampling instrument.

Response: Thanks, we had provided the detail of gas measuring (**Line 249-251**).

Comment 23: L246 superscript for -1

Response: Thanks, revised.

Comment 24: L272 'shown' rather than 'showed'

Response: Thanks, revised.

Comment 25: L285 Missing parentheses; dupliscated 'also'

Response: Thanks, revised.

We hope that you find our revision satisfactory. Thank you very much!

Respectfully,

Chengjie Ren, on behalf of all co-authors

College of Agronomy, Northwest A&F University, Yangling, 712100 Shaanxi, China;

The Research Center of Recycle Agricultural Engineering and Technology of Shaanxi Province, Yangling 712100 Shaanxi, China

Tel: +8613892872667, Fax: +86-87082104; Email: Rencj1991@nwsuaf.edu.cn

REVIEWER COMMENTS

Reviewer #1 (Remarks to the Author):

Thanks to the authors for addressing my concerns from the previous version of this manuscript. The authors have made valuable additions by including more CUE measurements at temperatures (5, 15, 25) and found a consistent variation of CUE along the MAT gradient. However, I still have some concerns about this manuscript.

The authors assume that CUE declines with increasing temperature due to microbial respiration and growth, both of which exponentially increase with temperature. However, if these variables rise at different rates with increasing temperature, the correlation between CUE and temperature should be non-linear.

It would be helpful if the authors could provide the R² and P values of the regression models used to simulate respiration/growth and temperatures. If the R² and P values of the regression models are low, it raises questions about the validity of the results obtained in this study.

In line 33-34, it is stated that the results indicate climate as the most important factor influencing CUE and CUEt. From Figure 3, I roughly calculated that soil C quality might also explain CUEt similar to climate. However, this should be double-checked.

In lines 36-38, it is mentioned that CUE decreases with temperature in cold regions and increases with temperature in warm regions. However, these patterns are difficult to discern in Figure 1a-c. I suggest changing the color gradients of the assay temperatures in Figure 1 to make the patterns more visible.

In lines 39-42, what is meant by "comfortable" climate conditions, and why do microbes reduce C loss in such conditions? From the results, we can observe a decline in CUE in cold regions, indicating increased C loss with higher temperatures, right?

In line 65, we can see that CUE does not always decrease with temperature, as demonstrated by labeling with Glucose and Oxalic acid (Frey et al. 2013 NCC).

In lines 113-116 and 331-339, it would be appropriate to conduct a multicollinearity analysis.

In lines 148-150, I could not find the equation for CUEt in the methods section. Please provide a graph showing the changes in CUE with assay temperature.

In lines 177-180, how could the microbial community be shifted under a 24 h incubation from bacteria- to fungal dominated?? This does not make sense.

In lines 205-217, this section is not easy to follow. What does SOC specific C assimilation mean? Growth plus respiration is referred to as microbial C uptake. Additionally, what do the green and pink points represent in Figure 4? Please provide additional calculations for these data. Lastly, how was the value of 24C derived?

Thanks to the authors for their thorough revisions of this manuscript. My concerns have been largely addressed, although the revisions have prompted a few additional comments:

Main comments

Framing of microbial CUE: while I appreciate the authors' more thorough presentation of the role of CUE in SOC retention in the introduction, I yet perceive that in the first paragraph and in the conclusions, the importance of CUE for SOC is still treated as a foregone conclusion when in my opinion the role of CUE for SOC has not been validated. The authors of this study cite the very recently published Tao et al. (ref 7), which is great, however a rebuttal to Tao et al. is currently available, although only in pre-print form (He et al., Contribution of carbon inputs to soil carbon accumulation cannot be neglected, doi: <https://doi.org/10.1101/2023.07.17.549330>). Of course, I'm not sure about the timing of this rebuttal relative to the authors' revisions and don't mean to imply they needed to have referenced earlier, or Nature's policy on citing pre-prints. I do think going forward it's important to acknowledge potential pitfalls of Tao et al.; the authors of the rebuttal argue that microbial CUE covaries with C inputs and that C inputs - which weren't included in the study of Tao et al. - are driving the spatial variability in SOC. Additionally, I extracted data from supplementary tables of Tao et al. to plot the raw (not log-transformed) relationship between SOC and CUE that go into Figure 1 of that paper. Although this approach doesn't account for within-site variability, as the Tao et al. did, it does show that global relationships between SOC and CUE are poorly resolved and seem driven by SOC observations above 50 g SOC / kg soil. I put these points forward to express concern about treating any strong relationship at all between SOC and CUE (either negative or positive) as a foregone conclusion, and to encourage the authors again to moderate throughout the text (including their conclusions).

Figure 1. Data from Tao et al. 2023 Figure 1, re-plotted by this reviewer to show data without log transformation.

Justification for linear rather than non-linear models for CUEt: The authors present Supplementary Table 3 of AIC values for both linear and non-linear models in response to a reviewer concern about whether the relationship between CUE and temperature is linear or non-linear. The authors claim in their response that these AIC values are lower for the linear models than the non-linear models, but it appears to me that the linear function has a higher AIC value for all reported sites. I am not sure how to reconcile this apparent disparity between SI Table 3, which seems to indicate non-linear models should be used to relate CUE to temperature, with the authors response, which seems to indicate that linear models were justified.

Grammar and wording throughout. While the writing is overall interpretable, unfortunately there are pervasive minor errors in word choice or grammar throughout the text. These are common enough they are beyond my capacity to provide specific suggestions for improvement. One example is L95, “This study aims to respond how do microbial CUE...”. This is not a correct formulation and could be replaced with ‘This study aims to describe how microbial CUE’ or ‘This study aims to characterize how microbial CUE’. Given the manuscript is at this stage of review and retains these issues, I would suggest to the author team to find assistance, if possible, to polish the word choice and grammar in each paragraph.

Minor comments

Paragraph starting on L79 and Figure 1 caption: replace ‘annul’ with ‘annual’

Figure 4. This figure stands out from preceding figures by not being clearly derived from the data reported in those figures. Where are these data and relationships coming from? Is this a conceptual figure? Was there process-based modeling done to support this? Please provide a clarifying sentence or two in the caption. If it’s a conceptual figure, suggest to start the caption with ‘Conceptual figure showing / based on / synthesizing...’

Dear Reviewers:

Thank you very much for considering the manuscript of *Thermal sensitivity of soil microbial carbon use efficiency across forest biomes*. We are grateful for the critical comments and suggestions raised by Reviewers on the manuscript, based on which we thoroughly revised it. Hope our revision relieved the concerns raised and enhanced the quality of the manuscript. For more details, please refer to point-by-point response to the reviewers' comments (The original Editors and Reviewers' comments are colored blue). In the revised manuscript, the revised parts have been shown in red ink text.

Reviewer #1 (Remarks to the Author):

Comment 1: Thanks to the authors for addressing my concerns from the previous version of this manuscript. The authors have made valuable additions by including more CUE measurements at temperatures (5, 15, 25) and found a consistent variation of CUE along the MAT gradient. However, I still have some concerns about this manuscript.

Response: We greatly appreciated your positive and constructive comments, which had substantially improved the manuscript.

Comment 2: The authors assume that CUE declines with increasing temperature due to microbial respiration and growth, both of which increase with temperature. However, if these variables rise at different rates with increasing temperature, the correlation between CUE and temperature should be non-linear.

Response: Thanks for your suggestions. First, we replaced the legend of “temperature” by “measuring temperature” in the Fig. 1a to be clearer (See Fig. 1a in revised manuscript).

Second, based on Akaike information criterion (AIC), we compared the performances of linear function vs exponential function in fitting the relationships between microbial physiological traits (growth, respiration, and CUE) and measuring temperature (5,10, 15, 20, 25, 30°C). We found that both microbial growth and respiration exponentially increase with increasing measuring temperature (**Supplementary Table 4**). However, linear function had consistent lower AIC values than exponential function in fitting the relationship between microbial CUE and measuring temperature across nine studied forests (**Supplementary Table 4**). Therefore, the slope of linear relationship between microbial CUE and measuring temperature is defined as the temperature sensitivity of microbial CUE (CUE_T).

Supplementary Table 4 Akaike information criterions for linear and exponential functions in fitting the relationships between microbial physiological traits and measuring temperature.

Microbial physiology	Study sites	Linear function	Exponential function
Carbon use efficiency	Mt. Maoer	-65.2473	-60.7426
	Mt. Dongling	-58.0453	-50.7939
	Fuxian	-62.8806	-50.0061
	Huoditang	-76.3749	-63.2925
	Maoxian	-57.5239	-53.5315
	Mt. Gongga	-78.6382	-72.5459
	Mt. Ailao	-92.6791	-85.8054
	Xishuangbanna	-58.148	-47.9978
	Jianfengling	-57.0253	-53.7097
Growth	Mt. Maoer	262.2253	253.3677
	Mt. Dongling	273.6244	271.0556
	Fuxian	260.6953	255.3485
	Huoditang	223.7732	216.4848
	Maoxian	229.6291	206.5959
	Mt. Gongga	217.2389	211.4432
	Mt. Ailao	209.7603	200.3569
	Xishuangbanna	198.3124	196.5032
	Jianfengling	208.7361	203.7676
Respiration	Mt. Maoer	245.8891	218.701
	Mt. Dongling	241.0022	228.7135
	Fuxian	248.1128	237.4673
	Huoditang	218.2446	191.1666
	Maoxian	227.672	218.9297
	Mt. Gongga	221.8945	217.3716
	Mt. Ailao	218.1439	208.5814
	Xishuangbanna	199.8814	197.786
	Jianfengling	208.6679	199.0898

Third, we further provided the scatter diagrams showing the relationships between microbial physiology traits and measuring temperature (5,10, 15, 20, 25, 30°C) in each forest (**showed in the following Fig. R1-R2**). In addition, the R^2 and P values were also provided in the scatter diagrams. **Fig. R3** clearly showed that microbial CUE linear decreased with measuring temperature in cold forests (Mt. Maoer, Mt. Dongling, Fuxian, Huoditang), while microbial CUE linear increased with measuring temperature in warm forest (Maoxian, Mt. Gongga, Mt. Ailao, Xishuangbanna, Jianfengling). That is to say, cold forest had a negative CUE_T , while warm forest had a positive CUE_T .

Fig. R1 Relationship between mass-specific respiration and measuring temperature in each forest. ME, Mt. Maoer; DL, Mt. Dongling; FX, Fuxian; HDT, Huoditang; MX, Maoxian; GG, Mt. Gongga; AL, Mt. Ailao; XSBN, Xishuangbanna; JFL, Jianfengling.

Fig. R2 Relationship between mass-specific growth and measuring temperature in each forest. ME, Mt. Maoer; DL, Mt. Dongling; FX, Fuxian; HDT, Huoditang; MX, Maoxian; GG, Mt. Gongga; AL, Mt. Ailao; XSBN, Xishuangbanna; JFL, Jianfengling.

Fig. R3 Relationship between microbial carbon use efficiency and measuring temperature in each forest. CUE, carbon use efficiency; ME, Mt. Maoer; DL, Mt. Dongling; FX, Fuxian; HDT, Huoditang; MX, Maoxian; GG, Mt. Gongga; AL, Mt. Ailao; XSBN, Xishuangbanna; JFL, Jianfengling.

Comment 3: It would be helpful if the authors could provide the R² and P values of the regression models used to simulate respiration/growth and temperatures. If the R² and P values of the regression models are low, it raises questions about the validity of the results obtained in this study.

Response: Thanks for your suggestions. We have provided the equations, R², and P values for the regressions in **Fig. 1a-c** (showed in **Supplementary Table 3**).

Supplementary Table 3 Relationships between microbial physiological traits and mean annual temperature at different measuring temperature. MAT, mean annual temperature; Mass-specific growth is microbial growth per unit microbial biomass carbon, while mass-specific respiration is microbial respiration per unit microbial biomass carbon.

Physiological traits	Measuring temperature (°C)	Equation	R ²	P
Carbon use efficiency	5	$y = -0.0185x + 0.7637$	0.82	<0.001
	10	$y = -0.0178x + 0.7560$	0.80	<0.001
	15	$y = -0.0134x + 0.68$	0.79	<0.001
	20	$y = -0.0144x + 0.7048$	0.81	<0.001
	25	$y = -0.0078x + 0.6323$	0.49	<0.001
	30	$y = -0.0091x + 0.6261$	0.68	<0.001
Mass-specific growth	5	$y = -0.0169x + 0.8216$	0.52	<0.001
	10	$y = -0.0398x + 1.3729$	0.44	<0.001
	15	$y = -0.0208x + 1.0935$	0.14	0.052
	20	$y = -0.0590x + 1.9619$	0.54	<0.001
	25	$y = -0.0387x + 2.1987$	0.13	0.061
	30	$y = -0.1087x + 4.0359$	0.39	<0.001
Mass-specific respiration	5	$y = 0.0250x + 0.2398$	0.69	<0.001
	10	$y = -0.0054x^2 + 0.1561x + 0.1424$	0.67	<0.001
	15	$y = 0.0260x + 0.4515$	0.35	<0.001
	20	$y = -0.0051x^2 + 0.1418x + 0.2984$	0.54	<0.001
	25	$y = 0.0200x + 1.1793$	0.11	0.091
	30	$y = -0.0101x^2 + 0.2538x + 1.2693$	0.44	<0.001

Comment 4: In line 33-34, it is stated that the results indicate climate as the most important factor influencing CUE and CUE_T. From Figure 3, I roughly calculated that soil C quality might also explain CUE_T similar to climate. However, this should be double-checked.

Response: We are agreed with you that soil C quality was also the important factor affecting microbial CUE and CUE_T except for MAT. In order to reduce confusing expression, revised as: Climate (mean annual temperature and precipitation) was the primary factor that negatively affected CUE and positively affected CUE_T, predominantly through direct pathways, then by altering soil properties, C fractions, microbial community structure, and functions.

Comment 5: In lines 36-38, it is mentioned that CUE decreases with temperature in cold regions and increases with temperature in warm regions. However, these patterns are difficult to discern in Figure 1a-c. I suggest changing the color gradients of the assay temperatures in Figure 1 to make the patterns more visible.

Response: Thanks for your suggestions. The temperature mentioned here (lines 38-39) is the measuring temperature rather than the mean annual temperature (MAT). In order to avoid such confusion, we replaced the legend of “temperature” by “measuring temperature” in the Fig. 1a (please refer to our response to your Comment 2).

In addition, we had showed the linear relationships between microbial CUE and measuring temperature in Fig. R3 (please refer to our response to your Comment 2).

Comment 6: In lines 39-42, what is meant by "comfortable" climate conditions, and why do microbes reduce C loss in such conditions? From the results, we can observe a decline in CUE in cold regions, indicating increased C loss with higher temperatures, right?

Response: Thanks for your suggestions. First, according to **Reviewer #2 Comment 5**, we had improved the quality of the **Fig. 4**. In specific, we added the boxplot indicate the microbial CUE and soil organic carbon (SOC)-specific C uptake (uptake/SOC), growth (growth/SOC), and C emission (respiration/SOC) of cold (had mean annual temperature lower than 10°C with negative temperature sensitivity of microbial CUE) vs warm (had mean annual temperature greater than 10°C with positive temperature sensitivity of microbial C use efficiency) forests at different measuring temperatures (5,10, 15, 20, 25, 30°C). We then fitted the temperature dependent uptake/SOC (exponential function), CUE (linear function), growth/SOC (exponential function), and C emission (exponential function) (**Fig. 4**).

Second, "comfortable" climate conditions for microbes in cold forests was low environmental temperature, while "comfortable" climate conditions for microbes in warm forests was high environmental temperature. Here, we found negative CUE_T in cold forest and positive CUE_T in warm forest. Therefore, the temperature optimum for microbial CUE may be lower in colder sites if organisms are adapted to the colder climate, while the temperature optimum for microbial CUE may be higher in warmer sites if organisms are adapted to the warmer climate (**Fig. 1**).

Third, we found that if temperature is lower than ~24°C, warm forests had greater C emission rate than cold forests at same temperature. However, if temperature is greater than ~24°C (temperature of the intersection point of two fitting curves), warm forests had lower C emission rate than cold forests at same temperature. Thus, climate-adaptive microbial community likely has a capacity to reduce C loss from soil matrix under corresponding "comfortable" climate conditions, plasticity of microbial CUE and its temperature sensitivity alter the feedback of soil C to climate warming.

Fig. 4 Regulation of microbial carbon use efficiency on soil carbon-climate feedbacks. Warm forests had mean annual temperature greater than 10°C with positive temperature sensitivity of microbial carbon (C) use efficiency (CUE) (red lines are the fitting values across all warm forests), while cold forests had mean annual temperature lower than 10°C with negative temperature sensitivity of microbial CUE (blue lines are the fitting values across all cold forests). Microbial C uptake is the sum of growth and respiration. SOC, soil organic C. The red and blue boxes are the microbial CUE and SOC-specific C uptake, growth, and respiration in warm and cold forests, respectively.

Comment 7: In line 65, we can see that CUE does not always decrease with temperature, as demonstrated by labeling with Glucose and Oxalic acid (Frey et al. 2013 NCC).

Response: Thanks for your suggestion. Using ^{13}C -labelled method, Frey et al. did find that temperature sensitivity of microbial CUE varied with the types of added-substrates. However, current microbial models assume a negative CUE_T due to the lack of experimental evidence of CUE_T using a substrate-independent approach. Here, our study found a negative CUE_T in cold forests, but a positive CUE_T in warm forests. Therefore, our findings revise the previous viewpoint in microbial models.

Comment 8: In lines 113-116 and 331-339, it would be appropriate to conduct a multicollinearity analysis

Response: Thanks for your suggestion. In our study, considering the correlations among factors within each group, we performed principal components analysis (PCA) within each group to create a new index for climates, soil properties, soil C quality, microbial community structure, and C decomposition genes (Qin et al. 2021). The first component, which accounted for $>60\%$ of the variance of each group (the selected

variables within each group was showed in **Supplementary Table 5**), was then introduced in the SEM. Thus, before conducting SEM, we have already conducted PCA analysis and removed the issue of multicollinearity.

Qin S, Kou D, Mao C, Chen Y, Chen L, Yang Y. Temperature sensitivity of permafrost carbon release mediated by mineral and microbial properties. *Science Advances*. 2021 Aug 6;7(32):eabe3596.

Comment 9: In lines 148-150, I could not find the equation for CUE_T in the methods section. Please provide a graph showing the changes in CUE with assay temperature.

Response: Thanks for your suggestion. We had evidenced the linear relationship between microbial CUE and measuring temperature (**Supplementary Table 3**). Therefore, the CUE_T is the slope of this linear relationship.

In the revised manuscript, we had clarified it in both “Results and Discussion” (Line 155-156) and “Method” (Line 314-315) sections.

Moreover, we have provided the linear relationship between microbial CUE and measuring temperature in **Figure R3**.

Comment 10: In lines 177-180, how could the microbial community be shifted under a 24 h incubation from bacteria- to fungal dominated?? This does not make sense

Response: Thanks for your suggestions, we have deleted these expression in revised manuscript.

Comment 11: In lines 205-217, this section is not easy to follow. What does SOC specific C assimilation mean? Growth plus respiration is referred to as microbial C uptake. Additionally, what do the green and pink points represent in Figure 4? Please provide additional calculations for these data. Lastly, how was the value of 24C derived?

Response: Thanks for your suggestion, we changed the “C assimilation” to “microbial C uptake”.

In **Fig. 4**, green points, solid pink points, and hollow pink points represent current condition, warming conditions altering microbial physiology, and warming conditions without altering microbial physiology, respectively. The curve is a fitting of the observed data (showed by boxplots). 24 °C is the intersection of two curve fitting point. More detail please refer to our **response to your Comment 6**.

Reviewer #2 (Remarks to the Author):

Comment 1: Framing of microbial CUE: while I appreciate the authors’ more through presentation of the role of CUE in SOC retention in the introduction, I yet perceive that in the first paragraph and in the conclusions, the importance of CUE for SOC is still treated as a foregone conclusion when in my opinion the role of CUE for SOC has not been validated. The authors of this study cite the very recently published Tao et al. (ref 7), which is great, however a rebuttal to Tao et al. is currently available, although only in pre-print form (He et al., Contribution of carbon inputs to soil carbon accumulation cannot be neglected, doi: <https://doi.org/10.1101/2023.07.17.549330>). Of course, I’m not sure about the timing of this rebuttal relative to the authors’ revisions and don’t

mean to imply they needed to have referenced earlier, or Nature’s policy on citing pre-prints. I do think going forward it’s important to acknowledge potential pitfalls of Tao et al.; the authors of the rebuttal argue that microbial CUE covaries with C inputs and that C inputs - which weren't included in the study of Tao et al. - are driving the spatial variability in SOC. Additionally, I extracted data from supplementary tables of Tao et al. to plot the raw (not log-transformed) relationship between SOC and CUE that go into Figure 1 of that paper. Although this approach doesn’t account for within-site variability, as the Tao et al. did, it does show that global relationships between SOC and CUE are poorly resolved and seem driven by SOC observations above 50 g SOC/kg soil. I put these points forward to express concern about treating any strong relationship at all between SOC and CUE (either negative or positive) as a foregone conclusion, and to encourage the authors again to moderate throughout the text (including their conclusions).

Figure 1. Data from Tao et al. 2023 Figure 1, re-plotted by this reviewer to show data without log transformation.

Response: Thanks for your insight suggestion on the contribution of microbial CUE to SOC. According to your comments, we had added another paragraph to discuss such potential pitfalls at the end of the manuscript (Line 217-230):

Our initial broad-scale exploration of microbial CUE_T highlights several future research needs regarding the regulation of microbial CUE for SOC storage. A recent study has suggested a positive contribution from microbial CUE (including microbial CUE calculated by various and incomparable methods) to global SOC storage (Tao et al., 2023). Indeed, we identified such positive correlations between microbial CUE and SOC consistently across six measurement temperatures (Supplementary Fig. 5). However, aside from a recent rebuttal based on statistical and process-based model structures considering C-inputs and C-quality (He et al., 2023), our findings verify that cold forests with high microbial CUE have a higher C emission rate (a more pronounced negative effect on SOC) if the temperature exceeds roughly 24°C (Fig. 4) compared to warmer forests with lower microbial CUE. Thus, the contribution of microbial CUE to SOC may be temperature dependent. In addition, we should quantify both microbial CUE and CUE_T in a more diverse range of ecosystems beyond the forests of China to provide more precise parameters for microbial models.

Ref1: Tao, F. *et al.* Microbial carbon use efficiency promotes global soil carbon storage. *Nature* **618**, 981–985 (2023).

Ref2: He, X.J., Abramoff, R., Abs, E., Georgiou, K., Zhang, H.C., Goll, D.S., Contribution of carbon inputs to soil carbon accumulation cannot be neglected. *bioRxiv* (2023). <https://doi.org/10.1101/2023.07.17.549330> (preprint)

Comment 2: Justification for linear rather than non-linear models for CUEt: The authors present Supplementary Table 3 of AIC values for both linear and non-linear models in response to a reviewer concern about whether the relationship between CUE and temperature is linear or nonlinear. The authors claim in their response that these AIC values are lower for the linear models than the non-linear models, but it appears to me that the linear function has a higher AIC value for all reported sites. I am not sure how to reconcile this apparent disparity between SI Table 3, which seems to indicate non-linear models should be used to relate CUE to temperature, with the authors response, which seems to indicate that linear models were justified.

Response: Thanks for your suggestions. We provided further evidence in the revised manuscript. Please refer to our responses to **Reviewer #1's Comment 2**.

Comment 3: Grammar and wording throughout. While the writing is overall interpretable, unfortunately there are pervasive minor errors in word choice or grammar throughout the text. These are common enough they are beyond my capacity to provide specific suggestions for improvement. One example is L95, “This study aims to respond how do microbial CUE...”. This is not a correct formulation and could be replaced with ‘This study aims to describe how microbial CUE’ or ‘This study aims to characterize how microbial CUE’. Given the manuscript is at this stage of review and retains these issues, I would suggest to the author team to find assistance, if possible, to polish the word choice and grammar in each paragraph.

Response: Thanks for your suggestions, we have asked a native speaker (Russell Doughty, PhD, a research scientist, Norman, Oklahoma, USA) to polish the word choice and grammar, which was showed as followed:

Comment 4: Paragraph starting on L79 and Figure 1 caption: replace ‘annul’ with ‘annual’

Response: Thanks, corrected.

Comment 5: Figure 4. This figure stands out from preceding figures by not being clearly derived from the data reported in those figures. Where are these data and relationships coming from? Is this a conceptual figure? Was there process-based modeling done to support this? Please provide a clarifying sentence or two in the caption. If it’s a conceptual figure, suggest to start the caption with ‘Conceptual figure showing / based on / synthesizing...’

Response: In our study, the **Fig 4** is based on the observed data, not a conceptual diagram. In order to make it clear, we have plotted the data distributions for the corresponding curve. More detail please refer to our **response to Reviewer #1's Comment 6**.

We hope that you find our revision satisfactory. Thank you very much!

Respectfully,

Chengjie Ren, on behalf of all co-authors

College of Agronomy, Northwest A&F University, Yangling, 712100 Shaanxi, China;

The Research Center of Recycle Agricultural Engineering and Technology of Shaanxi Province, Yangling 712100 Shaanxi, China

Tel: +8613892872667, Fax: +86-87082104; Email: Rencj1991@nwsuaf.edu.cn

REVIEWER COMMENTS

Reviewer #1 (Remarks to the Author):

I still have some concerns regarding the temperature sensitivity of CUE. From Fig. R3, it's noticeable that the regression R2 is low for the sites labeled GG and JFL. For the MX site, if we remove the two observations where CUE is higher than 0.549, there appears to be no correlation between the measuring temperature (T) and CUE. Similarly, for JFL, if we eliminate the outlier observation where CUE is lower than 0.30, there would be no change in CUE with measuring T. I believe the observed correlations for sites MX, GG, AL, XSBN, and JFL are too weak to be conclusive.

Furthermore, Fig.4 is challenging to comprehend. The authors failed to provide detailed explanations on how they calculated the microbial CUE response under the conditions of "without altering microbial physiology" and "altering microbial physiology".

Fig. S5 also seems to have some issues. The CUE was measured under different temperatures, but it corresponds to only one SOC value. It's implausible for the SOC to change so rapidly. Additionally, I believe all these regressions are not statistically significant at a P-value less than 0.05.

Reviewer #3 (Remarks to the Author):

Thanks to the authors for revising this manuscript.

One specific comment: Figure 4. The use of 10 C to separate 'warm forests' from 'cold forests' could be clarified within the figure itself by adding '> 10 C' (with degree symbol) and '< 10 C' to the current figure legend. Also suggest to clarify the x axis in this figure as 'measuring temperature' to correspond to previous figures.

Reviewer #4 (Remarks to the Author):

The authors did a thorough work considering the comments and suggestions by the reviewers in the revised version of the manuscript. I think they showed also by comparing linear vs. non-linear relationships using AIC seems to point out the best suiting model for each parameter. For CUE it seems that this is rather more linear, but for both growth and respiration the AIC seems to point rather to exponential fits (Table 4).

I think Fig R1-R3 are really helpful and also depict very clearly the linear vs. exponential relations and responses to the measuring temperature. However, it would be great to indicate here which sites are the 'cold' or 'warm' ones.

And actually it really made me think, if cooling a rather warm-adapted soil microbial community down to 0°C could be rather a shock and constrain them in both growth and respiration, while warming a cold adapted community might show the expected decreases in CUE.. and thereby affect the observed outcomes.

Additionally, I do have some smaller comments that could still be considered.

Line 47: actually I would suggest to phrase this statement the other way around; microbial CUE is the 'result' of C taken up by microbes allocated to respiration (i.e. released again from soil) to growth (potentially forming biomass and subsequently necromass, that could potentially form SOM, if it is not decomposed). So in models CUE might be one parameter that is controlling the fluxes, but in real soils it results from different rates.

Line 57: this reads as if only one ¹³C labelled substrate approach is used, but there have been many different substrate been used (e.g. glucose, carbohydrate or C compound mixtures, leucine additions etc...). this could be phrased more general like – Many results on microbial CUE are based on incubation with ¹³C labeled substrates.. ... and another point here – the response of the microbial communities to a labile C substrate can vary a lot depending on the microbial community as well as on the substrate complexity.

Line 60: the motivation should be clearly here to get a better, estimate of microbial physiology and temperature sensitivity of microbial communities across a large geographical gradient (and maybe not to finally use a novel method across a large gradient).

Line 74: I think it would be great here to also add what those studies found? What pattern emerged? Based on ¹³C tracing, what did predict microbial temperature sensitivity? This could point out more the need for a substrate independent method...?

Line 77: Fully agree here, more information is needed. From a soil/ecosystem model perspective sure CUE and CUET makes a lot of sense, but from a more soil focused perspective the controls over growth and respiration, respectively are also very interesting.

Line 124: To my knowledge there have more studies already shown that latitude or MAT or climate has a negative influence on microbial CUE (see e.g. Takriti et al 2018, or also Cruz-Paredes and Rousk 2024 and more), and this pattern emerges even with different, C substrate based methods.

Line 127: as you have both data on growth and respiration you can really not only suggesting here, but verify if this reduction in CUE was due to a stronger change of growth than respiration. However, still one cannot yet exclude also that the climate has an effect on plant C inputs that subsequently can affect C availability for microbial populations – or labile C inputs by plant roots for instance (e.g. depending on productivity).

Line 140: functional gene presence is not a really good predictor of real gene expression, and even subsequent enzyme activity rates, this data may be a bit overinterpreted here... it could also mean that those communities are far more efficient in internal compound recycling, or in microbial necromass recycling, and maybe contribute to accumulation of more complex C compounds.

Line 167: here at least a citation for fungal/bacterial differences could be considered as reference (see also references mentioned above).

Line 202: this is very interesting, and also kind of expectable, if the SOC is kind of a proxy for the amount of microbial biomass pool overall. However, I have a question to figure 4. Where do the two scenarios (orange and green points) derive from? This is not very intuitive and might need a better explanation.

Line 214: I think 'comfortable' climate conditions is not a very good term to use. Also from the graphs and figures I am not totally sure where/how exactly the 24 °C threshold was detected?

Line 249: From the methods it is not totally clear whether the samples used for CUE have been frozen after sampling? This I think could be very critical to clarify, as freezing/thawing can have a very strong impact on the active microbial community fraction, as well as on substrate availability and composition (e.g. lysing microbial cells, breaking aggregates).

Dear Reviewers:

Thank you very much for considering the manuscript of *Thermal sensitivity of soil microbial carbon use efficiency across forest biomes*. We are grateful for the critical comments and suggestions raised by Reviewers on the manuscript, based on which we thoroughly revised it. Hope our revision relieved the concerns raised and enhanced the quality of the manuscript. For more details, please refer to point-by-point response to the reviewers' comments (The original Editors and Reviewers' comments are colored blue). In the revised manuscript, the revised parts have been shown in red-inked text.

Reviewer #1 (Remarks to the Author):

Comment 1: I still have some concerns regarding the temperature sensitivity of CUE. From Fig. R3, it's noticeable that the regression R2 is low for the sites labeled GG and JFL. For the MX site, if we remove the two observations where CUE is higher than 0.549, there appears to be no correlation between the measuring temperature (T) and CUE. Similarly, for JFL, if we eliminate the outlier observation where CUE is lower than 0.30, there would be no change in CUE with measuring T. I believe the observed correlations for sites MX, GG, AL, XSBN, and JFL are too weak to be conclusive.

Response: Thanks for your suggestion. First, despite we deleted the points concerned, our critical conclusion of cold forests generally with negative temperature sensitivity of microbial CUE while warm forests generally with positive one is maintained. Second, the insignificant slope between microbial CUE and measuring temperature is also critical findings of our study because we highlight the change of temperature sensitivity of microbial CUE from negative to positive with increasing mean annual temperature. Third, as a regional study, it is expected that there is greater variability, resulting from multiple environmental factors (climate, soil properties, substrates, microbial attributes, etc.). Thus, we accept such variations of microbial CUE, considering there is no alternation of conclusions.

Comment 2: Furthermore, Fig.4 is challenging to comprehend. The authors failed to provide detailed explanations on how they calculated the microbial CUE response under the conditions of "without altering microbial physiology" and "altering microbial physiology".

Response: Thanks for your suggestion. The "without altering microbial physiology" and "altering microbial physiology" are theoretical speculation based on our findings, which is also referring from Box 1 from Singh et al. 2010. The "without altering microbial physiology" represents that the temperature sensitivity of microbial physiology was not changed, while "altering microbial physiology" represents an adaption of microbial physiology to temperature warming. In addition, we had revised the caption of Fig. 4 to be clearer.

Singh, B. K., Bardgett, R. D., Smith, P., & Reay, D. S. Microorganisms and climate change: terrestrial feedbacks and mitigation options. *Nature Reviews Microbiology*, 8, 779-790 (2010).

Box 1 | Microorganisms, process rates and climate models

The relationship between global changes (altered temperature, carbon dioxide (CO₂) levels and precipitation) and the rate of processes such as denitrification and respiration can change according to the response of microbial communities. For example, a soil process (such as the decomposition of organic carbon) converts a component from state 1 to state 2 at a rate k , and it is assumed that the process is mediated by the soil biota present (see the figure, part **a**). In the first scenario (see the figure, part **b**), global change directly influences the functioning of existing microbial communities without altering the community structure. This may cause a shift in the process rate, but its behaviour and controls remain unchanged. However, as in the second scenario (see the figure, part **c**), a shift in microbial community structure caused by global change could also alter the fundamental control mechanism of the process. Most ecosystem models and all climate models that include a description of microbial processes use first-order rate kinetics, which assume that the microbial population is sufficient to carry out the function (for example, decomposition) and that the rate of the process is modified by environmental factors such as temperature and moisture. This approach works well within the parameterized limits of the model, and process rates largely follow trajectories that are mimicked well by such formulations. What is not known, however, is what happens if the climate changes beyond the parameterized limits. For example, if the structure of the microbial community changes in such a way that the function also changes, a discontinuity in the response may occur and the response could move to a different trajectory (see the figure, part **c**). Such threshold effects cannot be represented in the current structure of ecosystem and coupled-climate models. Understanding these potential threshold effects and identifying the systems and processes for which they are likely to be of greatest importance remain key challenges for microbiology.

Figure part **a** is modified, with permission, from REF. 34 © (1998) Wiley and Sons, and figure parts **b** and **c** are modified, with permission, from REF. 13 © (1998) Wiley and Sons.

Comment 3: Fig. S5 also seems to have some issues. The CUE was measured under different temperatures, but it corresponds to only one SOC value. It's implausible for the SOC to change so rapidly. Additionally, I believe all these regressions are not statistically significant at a P-value less than 0.05.

Response: Thanks for your comment. Yes, it's impossible for the change of SOC during the short-term measurement of microbial CUE. However, in Fig. S5, the SOC value is constant, while microbial CUE is changing with measuring temperatures. In addition, the following **table** also showed the detail of the regressions.

Measuring temperatures (°C)	R^2	P
5	0.47	< 0.001
10	0.36	< 0.001
15	0.40	< 0.001
20	0.41	< 0.001
25	0.33	0.002
30	0.19	0.024

Reviewer #3 (Remarks to the Author):

Comment 1: Thanks to the authors for revising this manuscript.

Response: We greatly appreciated your positive comments.

Comment 2: One specific comment: Figure 4. The use of 10 C to separate ‘warm forests’ from ‘cold forests’ could be clarified within the figure itself by adding ‘> 10 C’ (with degree symbol) and ‘< 10 C’ to the current figure legend. Also suggest to clarify the x axis in this figure as ‘measuring temperature’ to correspond to previous figures.

Response: Thanks for your suggestion, and we have revised the Fig. 4.

Reviewer #4 (Remarks to the Author):

Comment 1: The authors did a thorough work considering the comments and suggestions by the reviewers in the revised version of the manuscript. I think they showed also by comparing linear vs. non-linear relationships using AIC seems to point

out the best suiting model for each parameter. For CUE it seems that this is rather more linear, but for both growth and respiration the AIC seems to point rather to exponential fits (Table 4). I think Fig R1-R3 are really helpful and also depict very clearly the linear vs. exponential relations and responses to the measuring temperature. However, it would be great to indicate here which sites are the ‘cold’ or ‘warm’ ones

Response: We greatly appreciated your positive comments. In our study, to be much clearer in presenting our critical conclusion, we do use the classification of cold and warm forests. The classification was based on mean annual temperature and the value of temperature sensitivity of microbial CUE. In specific, warm forests have a mean annual temperature greater than 10°C and a positive temperature sensitivity of microbial CUE, while cold forests have a mean annual temperature lower than 10°C and a negative temperature sensitivity of microbial CUE. We clarified it in abstract (**Line 41-43**) and main body (**line 584-588**).

Comment 2: And actually it really made me think, if cooling a rather warm-adapted soil microbial community down to 0°C could be rather a shock and constrain them in both growth and respiration, while warming a cold adapted community might show the expected decreases in CUE. and thereby affect the observed outcomes.

Response: Yes, this is the innovation of our article. Thank you very much for your recognition.

Comment 3: Additionally, I do have some smaller comments that could still be considered.

Response: We greatly appreciated your positive and constructive comments, which had substantially improved the manuscript.

Comment 4: Line 47: actually I would suggest to phrase this statement the other way around; microbial CUE is the ‘result’ of C taken up by microbes allocated to respiration (i.e. released again from soil) to growth (potentially forming biomass and subsequently necromass, that could potentially form SOM, if it is not decomposed). So in models CUE might be one parameter that is controlling the fluxes, but in real soils it results from different rates.

Response: Thanks for your suggestion, we have phrased this statement (**line 49-52**).

Comment 5: Line 57: this reads as if only one ¹³C labelled substrate approach is used, but there have been many different substrate been used (e.g. glucose, carbohydrate or C compound mixtures, leucine additions etc...). this could be phrased more general like – Many results on microbial CUE are based on incubation with ¹³C labeled substrates.. ... and another point here – the response of the microbial communities to a labile C substrate can vary a lot depending on the microbial community as well as on the substrate complexity.

Response: Thanks for your suggestion, we have phrased this statement (**line 60-65**).

Comment 6: Line 60: the motivation should be clearly here to get a better, estimate of microbial physiology and temperature sensitivity of microbial communities across a large geographical gradient (and maybe not to finally use a novel method across a large gradient).

Response: Thanks for your suggestion. According to your **Comment 5**, methods do introduce the noise. Therefore, it is urgent to use the novel method to improve our understanding of microbial CUE in regulating soil C cycling. We also rephrased the last sentence (**line 60-65**).

Comment 7: Line 74: I think it would be great here to also add what those studies found? What pattern emerged? Based on ¹³C tracing, what did predict microbial temperature sensitivity? This could point out more the need for a substrate independent method...?

Response: Thanks for your suggestion. First, we had added the current understanding of temperature sensitivity of respiration (**line 83-85**). Second, we had review of the findings of ¹³C tracing (However, existing experimental studies have not reported robust patterns; they have shown varied responses such as positive, negative, or no change with temperature based on the ¹³C-labeled substrate approach) (**line 78-80**).

Comment 8: Line 77: Fully agree here, more information is needed. From a soil/ecosystem model perspective sure CUE and CUET makes a lot of sense, but from a more soil focused perspective the controls over growth and respiration, respectively are also very interesting.

Response: We greatly appreciated your positive comments.

Comment 9: Line 124: To my knowledge there have more studies already shown that latitude or MAT or climate has a negative influence on microbial CUE (see e.g. Takriti et al 2018, or also Cruz-Paredes and Rousk 2024 and more), and this pattern emerges even with different, C substrate based methods.

Response: Thanks for your suggestion. We have deleted the expression of “the first to”.

Comment 10: Line 127: as you have both data on growth and respiration you can really not only suggesting here, but verify if this reduction in CUE was due to a stronger change of growth than respiration. However, still one cannot yet exclude also that the climate has an effect on plant C inputs that subsequently can affect C availability for microbial populations – or labile C inputs by plant roots for instance (e.g. depending on productivity).

Response: Thanks for your suggestion, we had compared the relationship between microbial CUE and growth and the relationship between microbial CUE and respiration, which directly supports that our original conclusion (Supplementary Fig. 3 and 4).

Additionally, we accepted your suggestion that climate may regulate the microbial CUE indirectly by altering forest structure, productivity, and other properties despite we did not measure these variables (**line 136-139**).

Supplementary Fig. 3 Relationship between microbial CUE and growth.

Supplementary Fig. 4 Relationship between microbial CUE and respiration.

Comment 11: Line 140: functional gene presence is not a really good predictor of real gene expression, and even subsequent enzyme activity rates, this data may be a bit overinterpreted here... it could also mean that those communities are far more efficient in internal compound recycling, or in microbial necromass recycling, and maybe contribute to accumulation of more complex C compounds.

Response: Thanks for your suggestion. Revised (line 159-170).

Comment 12: Line 167: here at least a citation for fungal/bacterial differences could be considered as reference (see also references mentioned above).

Response: Thanks for your suggestion, we have changed the citation for microbial (fungal and bacterial) differences for CUE (**line 189**).

Comment 13: Line 202: this is very interesting, and also kind of expectable, if the SOC is kind of a proxy for the amount of microbial biomass pool overall. However, I have a question to figure 4. Where do the two scenarios (orange and green points) derive from? This is not very intuitive and might need a better explanation.

Response: Thanks for your suggestion. More detail please refer to our responses to **Reviewer #1's Comment 2** and **line 639-652**.

Comment 14: Line 214: I think 'comfortable' climate conditions is not a very good term to use. Also from the graphs and figures I am not totally sure where/how exactly the 24 °C threshold was detected?

Response: Thanks for your suggestion. We have changed the "comfortable" to "favorable". **The intersection point of two fitting curves for the relationships between C emission and measuring temperature in warm and cold forests with a measuring temperature of ~24 °C.** Overall, we revised the caption of Fig. 4 (**line 639-652**).

Comment 15: Line 249: From the methods it is not totally clear whether the samples used for CUE have been frozen after sampling? This I think could be very critical to clarify, as freezing/thawing can have a very strong impact on the active microbial community fraction, as well as on substrate availability and composition (e.g. lysing microbial cells, breaking aggregates).

Response: In our study, in order to reduce the effects of freezing/thawing on soil microbial communities, a 7-day pre-incubation period in darkness at 25°C was conducted before isotope labeling.

We hope that you find our revision satisfactory. Thank you very much!